# Hydrogen peroxide positively regulates brassinosteroid signaling through oxidation of the BRASSINAZOLE-RESISTANT1 transcription factor

Yanchen Tian[1], Min Fan[1], Zhaoxia Qin[1], Hongjun Lv[1], Minmin Wang[1], Zhe Zhang[2], Wenying Zhou[1], Na Zhao[1], Xiaohui Li[1], Chao Han[1], Zhaojun Ding[1], Wenfei Wang[2], Zhi-Yong Wang [3] Ming-Yi Bai[1]

Hydrogen peroxide ($H_2O_2$) is an important signaling molecule in plant developmental processes and stress responses. However, whether $H_2O_2$-mediated signaling crosstalks with plant hormone signaling is largely unclear. Here, we show that $H_2O_2$ induces the oxidation of the BRASSINAZOLE-RESISTANT1 (BZR1) transcription factor, which functions as a master regulator of brassinosteroid (BR) signaling. Oxidative modification enhances BZR1 transcriptional activity by promoting its interaction with key regulators in the auxin-signaling and light-signaling pathways, including AUXIN RESPONSE FACTOR6 (ARF6) and PHYTO-CHROME INTERACTING FACTOR4 (PIF4). Genome-wide analysis shows that $H_2O_2$-dependent regulation of BZR1 activity plays a major role in modifying gene expression related to several BR-mediated biological processes. Furthermore, we show that the thioredoxin TRXh5 can interact with BZR1 and catalyzes its reduction. We conclude that reversible oxidation of BZR1 connects $H_2O_2$-mediated and thioredoxin-mediated redox signaling to BR signaling to regulate plant development.

[1] Key Laboratory of Plant Development and Environment Adaptation Biology, Ministry of Education, School of Life Sciences, Shandong University, Jinan, China. [2] Basic Forestry and Proteomics Research Center, Fujian Agriculture and Forestry University, Fuzhou, China. [3] Department of Plant Biology, Carnegie Institution for Science, Stanford, CA, USA. These authors contributed equally: Yanchen Tian, Min Fan, Zhaoxia Qin. Correspondence and requests for materials should be addressed to M.-Y.B. (email: baimingyi@sdu.edu.cn)

Reactive oxygen species (ROS) are initially considered to be toxic by-products that accumulate under stress conditions and cause irreversible damage to cells[1]. Recently, ROS are recognized as important signaling molecules that regulate normal plant growth and development, as well as stress responses[1,2]. ROS include superoxide anion ($O_2^{\bullet-}$), hydrogen peroxide ($H_2O_2$), singlet oxygen ($^1O_2$), and hydroxyl radical ($OH^{\bullet}$). Among these, $H_2O_2$ is considered an important redox signaling molecule, given its specific physical and chemical properties, including a remarkable stability within cells (half time of $10^{-3}$ s), and rapid and reversible oxidation of target proteins[3]. $H_2O_2$ can cause oxidation of thiol groups of cysteines in target proteins, which either activates, inactivates, or alters their structure and function[3]. Elucidation of how $H_2O_2$ is sensed and how this signal is transmitted to the cell machinery is central to our understanding of so-called "redox signaling" pathways.

$H_2O_2$ interplays with diverse phytohormone signals to regulate plant developmental processes and stress responses[4]. Abscisic acid is a stress hormone that has important roles in regulating plant responses to abiotic stress as well as controlling seed germination and stomatal movement. ABA treatment induces the production of $H_2O_2$, which in turn activates the plasma membrane calcium channel and promotes the stomatal closure. ABA-insensitive loci ABI1 and ABI2, which encode protein phosphatase 2C, are the key negative components of the ABA signaling pathway, in which $H_2O_2$ may directly inhibit PP activity of ABI1 and ABI2 to positively regulate the transduction of ABA[5,6]. Salicylic acid (SA) plays crucial roles in the plant defense response to biotrophic pathogens. Upon exposure to biotrophic pathogens, SA accumulates and triggers an early burst of $H_2O_2$ by binding to CATALASE2 (CAT2), thereby inhibiting catalysis of $H_2O_2$ decomposition to water and oxygen. An increasing level of $H_2O_2$ induces the sulfenylation of the auxin-biosynthesis enzyme, tryptophan synthetase B subunit 1 and subsequently reduces the auxin levels. At the same time, SA binding to CAT2 also inhibits the stimulation of the enzymatic activity of jasmonic acid (JA) biosynthesis genes ACYL-CoA OXIDASE 2 and 3 (ACX2 and ACX3) to repress JA production. The decrease in auxin and JA accumulation reinforces plant immunity against biotrophic pathogens[7]. Following the SA-induced increase in oxidation potential, which did not last long, plant cells overcompensate with both enzymatic and non-enzymatic antioxidants to protect against oxidative stress. NONEXPRESSOR of PR GENES 1 (NPR1) senses the SA-mediated cellular redox changes and undergoes a change from the oligomeric state to the monomeric form. The redox-activated NPR1 is translocated to the nucleus, where it interacts with TGA transcription factors to activate the expression of PR genes and the plant immune response[8].

Brassinosteroids (BRs) are a group of plant steroid hormones that play fundamental roles in plant growth, development, and stress responses[9,10]. BR perception through the plasma membrane-localized receptor kinase BRASSINOSTEROID INSENSITIVE1 (BRI1) promotes the association of BRI1 with co-receptor BRI1-ASSOCIATED KINASE1 (BAK1) and induces a phosphorylation-dephosphorylation cascade that activates BRASSINAZOLE-RESISTANT1 (BZR1) and BRI1-EMS-SUPPSSOR1 (BES1), two homologous transcription factors, to regulate expression of thousands of BR-responsive genes[11,12]. In the absence of BR, BZR1 and BES1 are phosphorylated by the GSK3-like kinase BRASSINOSTEROID-INSENSITIVE2 (BIN2), they lose their DNA-binding activity, and are retained in the cytosol due to binding to the 14-3-3 proteins[12]. When BR levels are high, BZR1 is dephosphorylated by protein phosphatase 2A (PP2A), and subsequently translocated into the nucleus to bind and regulate target gene expression[13,14].

Recent studies uncovered a role of $H_2O_2$ in BR regulation of plant growth and stress responses[15,16]. Low BR levels induce a transient $H_2O_2$ production and change the cellular redox status in guard cells, thus resulting in stomata opening[15]. High BR levels induce prolonged $H_2O_2$ accumulation, which facilitates stress responses and stomata closure. Inhibition of $H_2O_2$ accumulation by chemical agents, such as ascorbic acid or diphenylene iodonium (DPI)[17], blocked BR-induced stomata closure, indicating an essential role for $H_2O_2$ in BR-induced stomata closure. However, the interplay between $H_2O_2$ and BR signaling in plant development remains unclear.

In this study, we demonstrate a critical role of $H_2O_2$ in BR regulation of gene expression and seedling growth and development. We find that BR not only inhibits the kinase activity of BIN2 to promote BZR1 dephosphorylation and DNA-binding ability, but also triggers the accumulation of $H_2O_2$ to induce BZR1 oxidation, which increases the transcriptional activity of BZR1 by enhancing its binding affinity to its partners such as PIF4 and ARF6. We further show that oxidized BZR1 is reduced by a thioredoxin protein, TRXh5. Mutations of oxidative sites in BZR1 and overexpression of TRXh5 both reduce BZR1 functions in cell elongation and QC cell division. Our study demonstrates that BZR1 activity is regulated by the opposing redox actions of $H_2O_2$ and TRXh5.

## Results

**$H_2O_2$ is required for the BR-mediated seedling development.** To determine whether $H_2O_2$ is necessary for BR function during Arabidopsis seedling development, we measured $H_2O_2$ content in the presence or absence of BR using the amplex red hydrogen peroxide/peroxidase assay. The results showed that BR treatment significantly increased the $H_2O_2$ level in the seedlings (Supplementary Fig. 1a). Similarly, analysis of $H_2O_2$ level by the fluorescent dye 2′,7′-dichlorofluorescin diacetate ($H_2$DCFDA)[18] showed BR induction of $H_2O_2$ in wild-type and BR-deficient mutant rot3-2, but not in the BR-insensitive mutant bri1-116, and this induction was impaired in the presence of the NADPH oxidase inhibitor, DPI[17] (Supplementary Fig. 1b–f). These results suggest that BR signaling through BRI1 triggers the production of $H_2O_2$ through a NADPH-dependent pathway.

To evaluate the potential roles of $H_2O_2$ in BR-mediated seedling development, we analyzed BR regulation of hypocotyl elongation under the $H_2O_2$-deficient condition. Our results showed that DPI treatment, which reduces the $H_2O_2$ levels in plants, significantly inhibited the effects of BR on hypocotyl elongation, with high concentrations of DPI causing seedlings insensitivity to BR (Fig. 1a). $H_2O_2$ effectively restored the BR sensitivity in DPI-treated plants in a dose-dependent manner (Fig. 1b), indicating that the accumulation of $H_2O_2$ is required for BR-induced hypocotyl elongation.

Because BR treatment increased $H_2O_2$ level in root tips, including the root stem cell niches (Supplementary Fig. 1b–d), we tested whether $H_2O_2$ plays an important role in BR-mediated root meristem development. In agreement with previous results[19,20], exogenous BR treatment increased the frequency of QC cell division about 6-fold, from 10 to 59%, in 5-day-old wild-type seedlings. Co-treatments with DPI and potassium iodide strongly counteracted the BR effects and reduced the frequency of QC division from 59 to 21 and 44%, respectively (Supplementary Fig. 2a), suggesting the production of $H_2O_2$ is necessary for BR promotion of QC cell division.

To corroborate these pharmacological data, we analyzed the effects of $H_2O_2$ on BR-induced cell elongation and QC cell division in the rbohDrbohF double mutant[21] and catalase-overexpressing plants (CAT2-Ox and CAT3-Ox)[22], which are

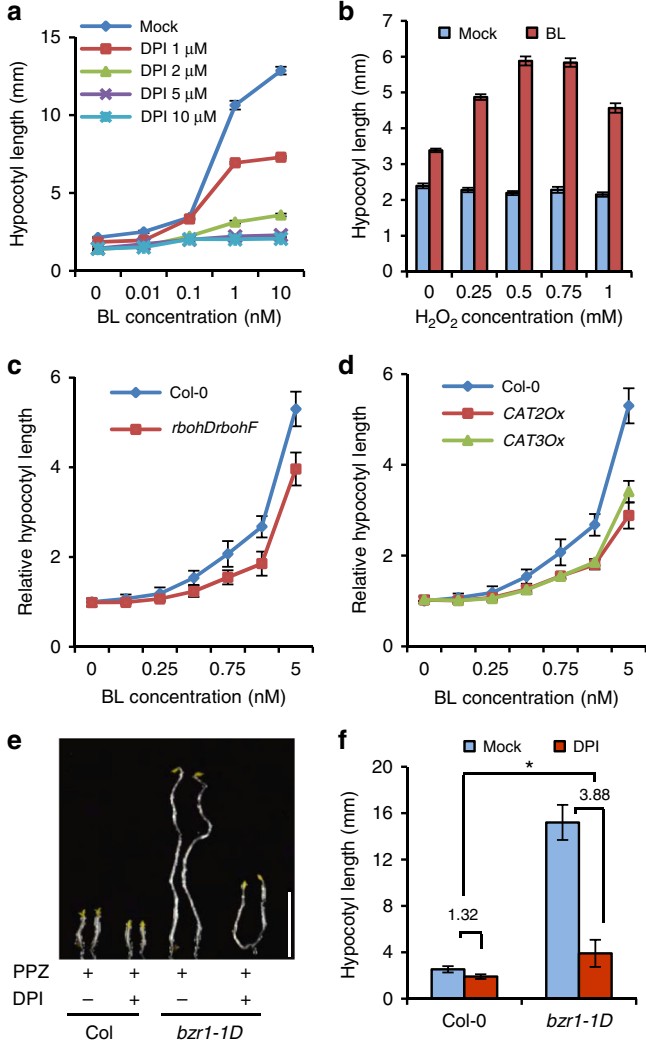

**Fig. 1** The production of $H_2O_2$ is required for BR and BZR1 promotion of cell elongation. **a** DPI treatment attenuated the plant sensitivity to BR. Wild-type (Col-0) seedlings were grown on ½ MS medium containing 2 µM PPZ and different concentrations of BL and DPI in the dark for 6 days. Error bars, s.d. ($n = 30$ plants). **b** $H_2O_2$ restored the BR sensitivity of DPI-treated plants. Wild-type Col-0 seedlings were grown on ½ MS medium containing 2 µM PPZ, 2 µM DPI, and different concentrations of $H_2O_2$, and with or without 1 nM BL in the dark for 6 days. Error bars, s.d. ($n = 30$ plants). **c, d** The rbohDrbohF mutant and catalase-overexpressing plants (CAT2-Ox and CAT3-Ox) showed BR hyposensitivity in hypocotyl elongation assay. Seedlings of Col-0, rbohDrohbF, CAT2-Ox, and CAT3-Ox were grown on ½ MS medium containing 2 µM PPZ and different concentrations of BL in the dark for 6 days. Error bars, s.d. ($n = 30$ plants). **e, f** DPI attenuated the PPZ-resistant phenotype of bzr1-1D mutant. Wild-type Col-0 and bzr1-1D seedlings were grown on ½ MS medium containing 2 µM PPZ and 0 or 1 µM DPI in the dark for 6 days. Scale bar, 5 mm. Error bars, s.d. ($n = 25$ plants). *$p < 0.05$, as determined by a Student's t-test

defective in normal production of $H_2O_2$. The results showed that rbohDrohbF, CAT2-Ox, and CAT3-Ox all showed BR hyposensitivity in hypocotyl elongation (Fig. 1c, d) and QC cell division (Supplementary Fig. 2b, c). Together, these results demonstrated that the BR-triggered production of $H_2O_2$ contributes to the BR-promoted cell elongation and QC cell division.

**$H_2O_2$ contributes to BR signaling by targeting BZR1.** The GSK3 kinase BIN2 is a key negative regulator in BR signal

transduction, and is involved in several other signaling pathways by interacting with and/or phosphorylating additional substrates[23]. The kinase activity of BIN2 was recently reported to be inhibited by nitric oxide in a dose-dependent manner[24]. $H_2O_2$ and NO sometimes share common target proteins to transduce redox signals. For example, NPR1, the master regulator in plant immunity, is induced to occur oxidative modification by both $H_2O_2$ and NO and then undergoes polymerization and loss of its activity[8,25]. To determine whether $H_2O_2$ regulates BR signal transduction through BIN2, we examined the effects of $H_2O_2$ on the cell elongation and QC cell division phenotypes in the bin2–3bil1bil2 mutant lacking BIN2 and its two homologs. The results showed that the bin2–3bil1bil2 mutant exhibited a long hypocotyl phenotype in the presence of the BR biosynthesis inhibitor propiconazole (PPZ), but this phenotype was suppressed by DPI treatment (Supplementary Fig. 3a). Microscopic observation showed that bin2–3bil1bil2 displayed higher frequency of QC cell division than wild-type seedlings. Both DPI and KI treatments significantly reduced the frequency of QC cell division in bin2–3bil1bil2 (Supplementary Fig. 3b). Considering BIN2 belongs to a GSK3 family containing 10 members in the Arabidopsis genome, and among which seven members act redundantly with BIN2 in the BR signaling pathway, two specific GSK3 kinase inhibitors, lithium chloride (LiCl) and bikinin[26,27], were used to block the kinase activity of GSK3 family proteins. The results showed that both LiCl and bikinin significantly increased the cell elongation and the frequency of QC cell division, as observed upon genetic or ligand-induced activation of the BR pathway. In contrast removal of $H_2O_2$ by DPI treatment decreased the hypocotyl length and the frequency of QC cell division, suggesting that involvement of $H_2O_2$ in the BR signaling pathway is likely independent of BIN2 and its homologous proteins (Supplementary Fig. 3c–f).

Further confirmation of the crosstalk between $H_2O_2$ and BR downstream of BIN2 was obtained by analyzing cell elongation and QC cell division in transgenic plants expressing BZR1ΔDM, a mutant BZR1 with deletion of the 12-amino-acid BIN2-docking motif resulting in the loss of BIN2 association and phosphorylation[28]. These plants exhibited longer hypocotyl and higher frequency of QC cell division than wild-type plants, while DPI treatment also inhibited these phenotypes (Supplementary Fig. 3g, h). Consistent with these results, DPI treatment reduced the cell elongation in the BR-activated mutants bzr1-1D and bes1-D mutants grown on the medium containing the BR biosynthesis inhibitor PPZ, which causes inactivation of wild-type BZR1 and BES1 but not bzr1-1D and bes1-D proteins (Fig. 1e, f and Supplementary Fig. 3i), suggesting that BZR1/BES1-driven cell elongation is abrogated by DPI treatment. DPI also reduced QC cell division in bzr1-1D and bes1-D (Supplementary Fig. 3j, k). Together, these genetic and physiological data suggest that the $H_2O_2$ regulates the BR activity downstream of BIN2, and probably through the transcription factors BZR1 and BES1.

**$H_2O_2$ induces the oxidative modification of BZR1.** The direct regulatory effects exerted by $H_2O_2$ are mediated by post-translational modification of target proteins through cysteine oxidative modification[29]. It has been reported that biotin-conjugated iodoacetamide (BIAM) and $H_2O_2$ could selectively and competitively react with cysteine residues that exhibit a low $pK_a$ in target proteins[30], and thus the $H_2O_2$-sensitive and $H_2O_2$-oxidized cysteine residues can be detected by BIAM labeling assay and biotin-switch assay, respectively (Fig. 2a, b). To test whether $H_2O_2$ induces oxidation of BZR1, maltose-binding protein (MBP) and MBP-BZR1 were pretreated with or without $H_2O_2$, and then incubated with BIAM. SDS-PAGE followed by immunoblot

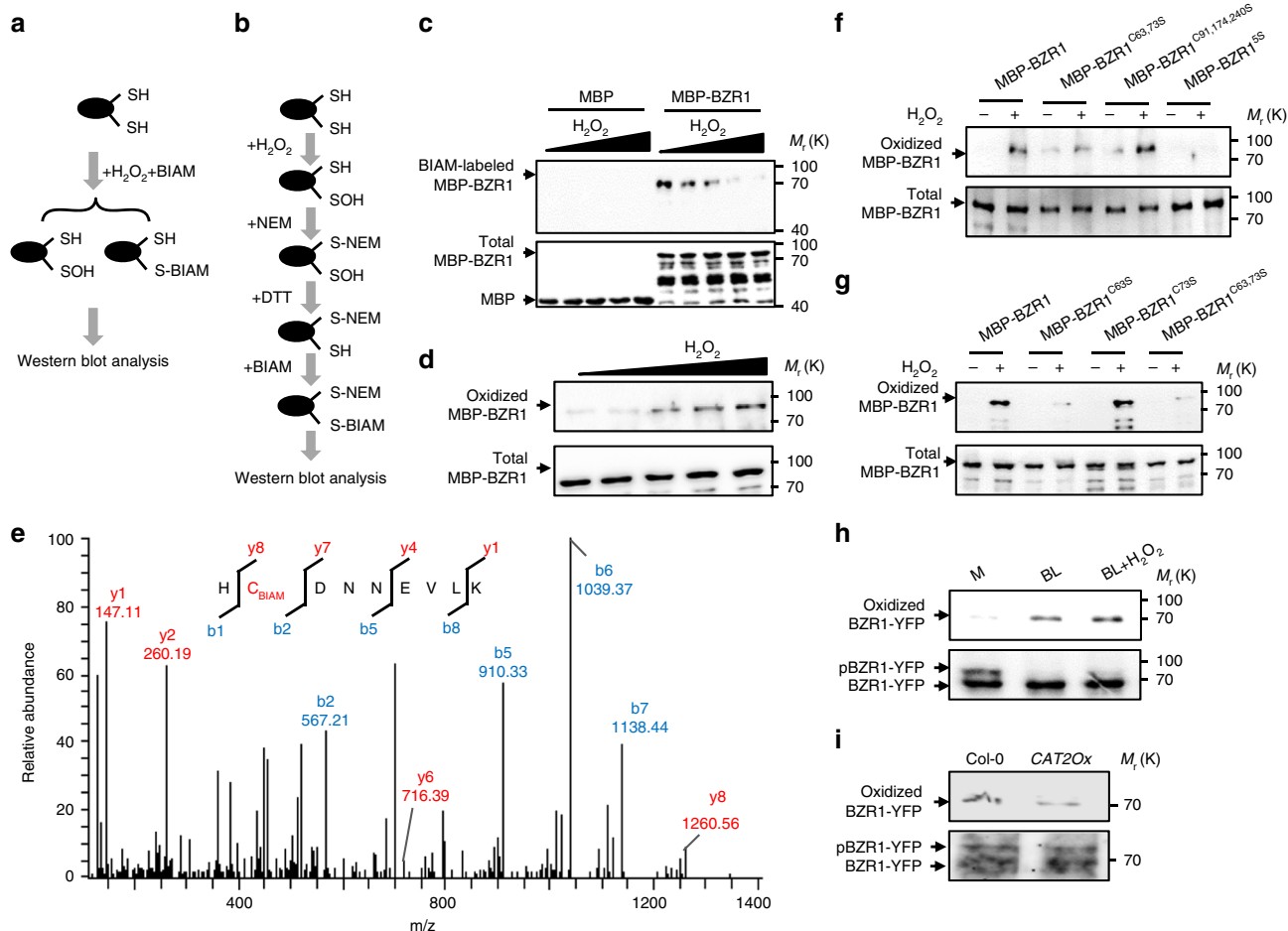

**Fig. 2** $H_2O_2$ induced the oxidation of BZR1 in vitro and in vivo. **a**, **b** The flowcharts show the procedures for quantifying redox modification of target proteins by BIAM-labeling assay (**a**) and biotin-switch assay (**b**). **c** Analysis of the oxidative modification of BZR1 by BIAM-labeling assay. MBP and MBP-BZR1 protein were pretreated with different concentrations of $H_2O_2$, and then directly labeled by BIAM at room temperature for 30 min. The labeled proteins were subjected to separation by SDS-PAGE, and detected by gel blot analysis with HRP-conjugated streptavidin and anti-MBP antibodies. **d** Analysis of the oxidative modification of BZR1 by the biotin switch assay. MBP-BZR1 proteins pretreated with different concentrations of $H_2O_2$ were first incubated with NEM to irreversibly block the free thiols, then treated with DTT to reduce the pre-existing oxidized cysteines in MBP-BZR1. The newly exposed free thiol groups were then relabeled with BIAM. The BIAM-tagged proteins in the samples were then captured with streptavidin beads and detected by gel blot using anti-MBP antibody. **e** Mass spectrometry analysis of the tryptic fragments of of MBP-BZR1 protein treated as in **d**. Cys-63 charged with BIAM was identified as an $H_2O_2$-sensitive residue. **f**, **g** Effects of mutating various cysteine residues on the $H_2O_2$-induced cysteine oxidation in MBP-BZR1, analyzed as in **d**. **h**, **i** BL and $H_2O_2$ treatment induced (**h**), but overexpression of *CAT2* (**i**) decreased the oxidative modification of BZR1 proteins in plants. Total proteins from *p35S:BZR1-YFP* transgenic plants treated with or without 100 nM BL and 1 mM $H_2O_2$ for 3 h, or from the *p35S:BZR1-YFP/p35S:CAT2* transgenic plants were sequentially treated with NEM, DTT, and BIAM, and then analyzed for biotin label

analysis detected the biotin-labeled proteins. As shown in Fig. 2c, MBP-BZR1, but not MBP only, was labeled by BIAM, while the labeling levels were decreased by $H_2O_2$ treatment in a dose-dependent manner, indicating that $H_2O_2$ decreased the reduced form of cysteine residues of BZR1. To further detect cysteine oxidation, $H_2O_2$-treated MBP-BZR1 was first incubated with a thiol alkylation reagent, N-ethylmaleimide (NEM), which irreversibly modifies free thiols in proteins. The NEM-modified MBP-BZR1 was then precipitated, treated with DTT to reduce any oxidized cysteines pre-existing before NEM treatment. The newly exposed free thiol groups were then labeled with BIAM. The BIAM-tagged proteins in the samples were then immuno-precipitated with streptavidin beads and revealed by western blot analysis. The results show that $H_2O_2$ causes oxidation of BZR1 in vitro (Fig. 2d).

Five cysteine residues in BZR1 are putative target sites of oxidation, two of them, cys-63 and cys-73, are located in the BZR1 N-terminal DNA-binding domain, while cys-91, cys-174,

and cys-240, are located in the BZR1 C-terminal phosphorylation domain (Supplementary Fig. 4a, b). To investigate which cysteine sites of BZR1 are oxidized by $H_2O_2$, we analyzed tryptic fragments derived from recombinant BZR1 protein sequentially treated with $H_2O_2$, NEM, and BIAM by liquid chromatography-tandem mass spectrometry, and identified cys-63 (Fig. 2e), cys-91, cys-174, and cys-240 (Supplementary Fig. 5a–c) as oxidized residues. Then, we mutated each cysteine (C) to serine (S) to determine which cysteine residues are the major oxidized sites in BZR1. Simultaneous mutagenesis of cys-91, cys-174, and cys-240 had no significant effect on oxidation of BZR1, while simultaneous mutagenesis of cys-63 and cys-73 significantly decreased BZR1 oxidation (Fig. 2f). Further experiments showed that MBP-BZR1$^{C73S}$ exhibited similar oxidation levels with wild-type MBP-BZR1, but MBP-BZR1$^{C63S}$ showed a very weak oxidative modification (Fig. 2g), suggesting the $H_2O_2$-induced oxidation occurs mainly on cys-63 of BZR1. Moreover, we showed that mutagenesis of cys-84 in BES1, which is equivalent to the

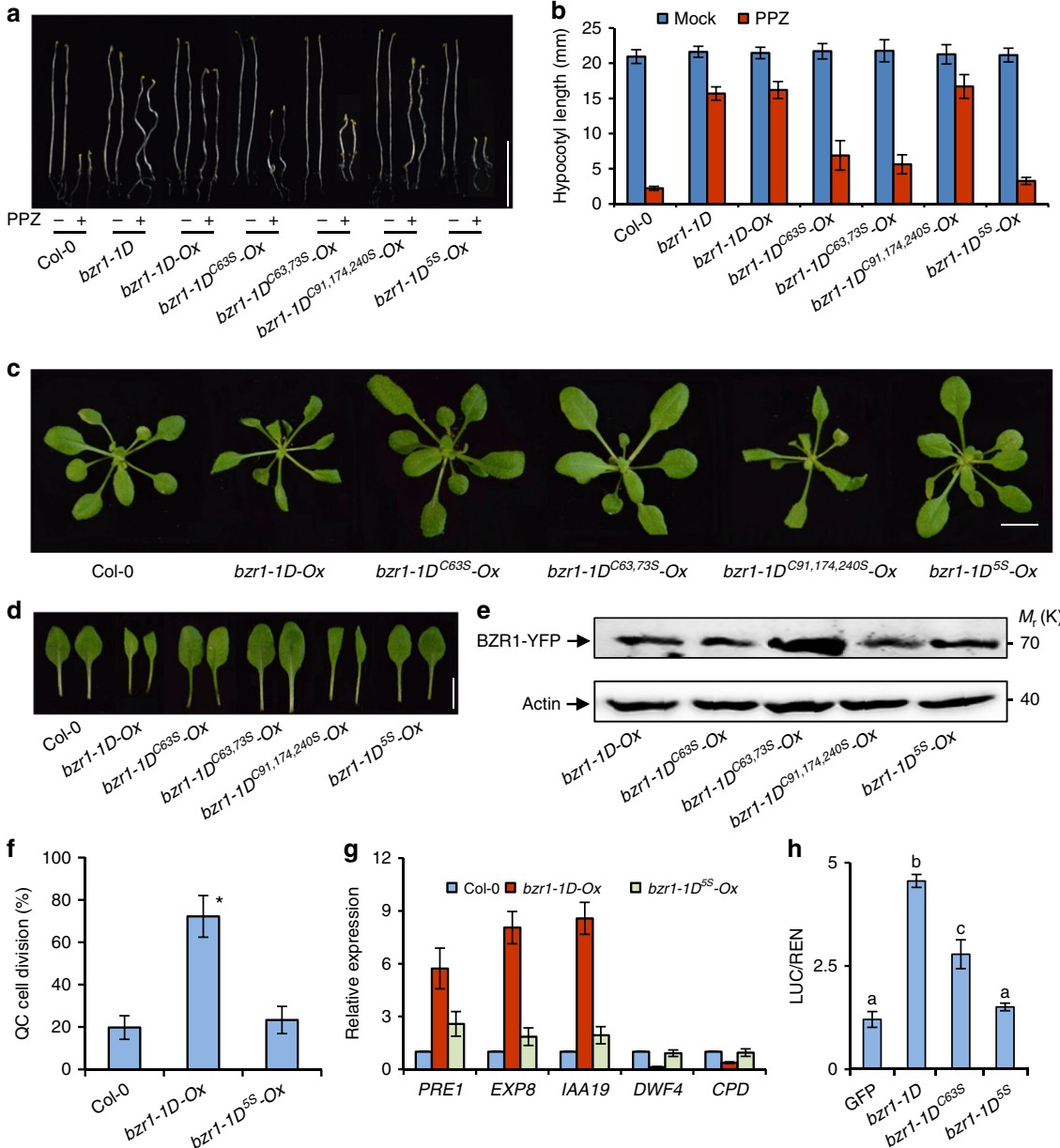

**Fig. 3** Mutations of the oxidized cysteines reduce BZR1 activity in transgenic plants. **a**, **b** The *bzr1-1D^C63S^*, *bzr1-1D^C63,73S^*, and *bzr1-1D^5S^* mutations attenuated the PPZ-resistant phenotype of *bzr1-1D*. Seedlings were grown in the dark on ½ MS medium containing mock solution or 2 μM PPZ for 6 days, and the hypocotyl length of each seedling was measured. Scale bar, 10 mm. Error bars indicated s.d. (*n* = 30 plants). **c**, **d** Phenotype of transgenic plants expressing mutated *BZR1* grown in soil for 3 weeks under long-day condition. Scale bar, 10 mm. **e** Immunoblot analysis of the protein levels of BZR1 in different transgenic plants using anti-YFP antibody. An antibody against actin was used to verify equal protein loadings. **f** The *bzr1-1D^5S^* mutation suppressed the BZR1-induced QC cell division. (*t*-test; *\*p* < 0.05). **g** Quantitative RT-PCR analysis of gene expression in wild-type, *bzr1-1D-Ox*, and *bzr1-1D^5S^-Ox* seedlings. *PP2A* was used as the internal control. **h** The *bzr1-1D^C63S^* and *bzr1-1D^5S^* mutation impaired the transcriptional activity of BZR1 in protoplast. Different letters above the bars indicate statistically significant differences between the samples (two-way ANOVA, *p* < 0.05). Error bars of **f**–**h** indicated standard deviation from three biological repeats

conserved residue cys-63 in BZR1 protein, effectively prevented the oxidation of BES1 by $H_2O_2$ treatment, suggesting that the conserved cys-63 and cys-84 are the major in vitro oxidized sites for BZR1 and BES1, respectively (Supplementary Figs. 4b, 6a, b). Finally, to determine whether BZR1 is oxidatively modified in vivo, total protein was extracted from the *p35S:BZR1-YFP* transgenic seedlings pretreated with or without BL and $H_2O_2$. The biotin-switch assay showed that BR treatment clearly induced the oxidation of BZR1, and application of $H_2O_2$ enhanced the oxidative effect (Fig. 2h). In addition, we found that over-expression of *CAT2* significantly reduced the oxidative

modification of BZR1 (Fig. 2i). These results suggest that $H_2O_2$ regulates BZR1 redox state by oxidation.

**Cys-63 is important for the functions of BZR1**. To understand whether oxidation of cys-63 modulates the function of BZR1 in plants, we generated transgenic Arabidopsis plants expressing $BZR1^{P234L}$ (*bzr1-1D*), $BZR1^{P234LC63S}$ (*bzr1-1D^C63S^*), $BZR1^{P234LC63,73S}$ (*bzr1-1D^C63,73S^*), $BZR1^{P234LC91,174,240S}$ (*bzr1-1D^C91,174,240S^*), and $BZR1^{P234LC63,73,91,174,240S}$ (*bzr1-1D^5S^*). Transgenic lines expressing similar protein levels of BZR1 were

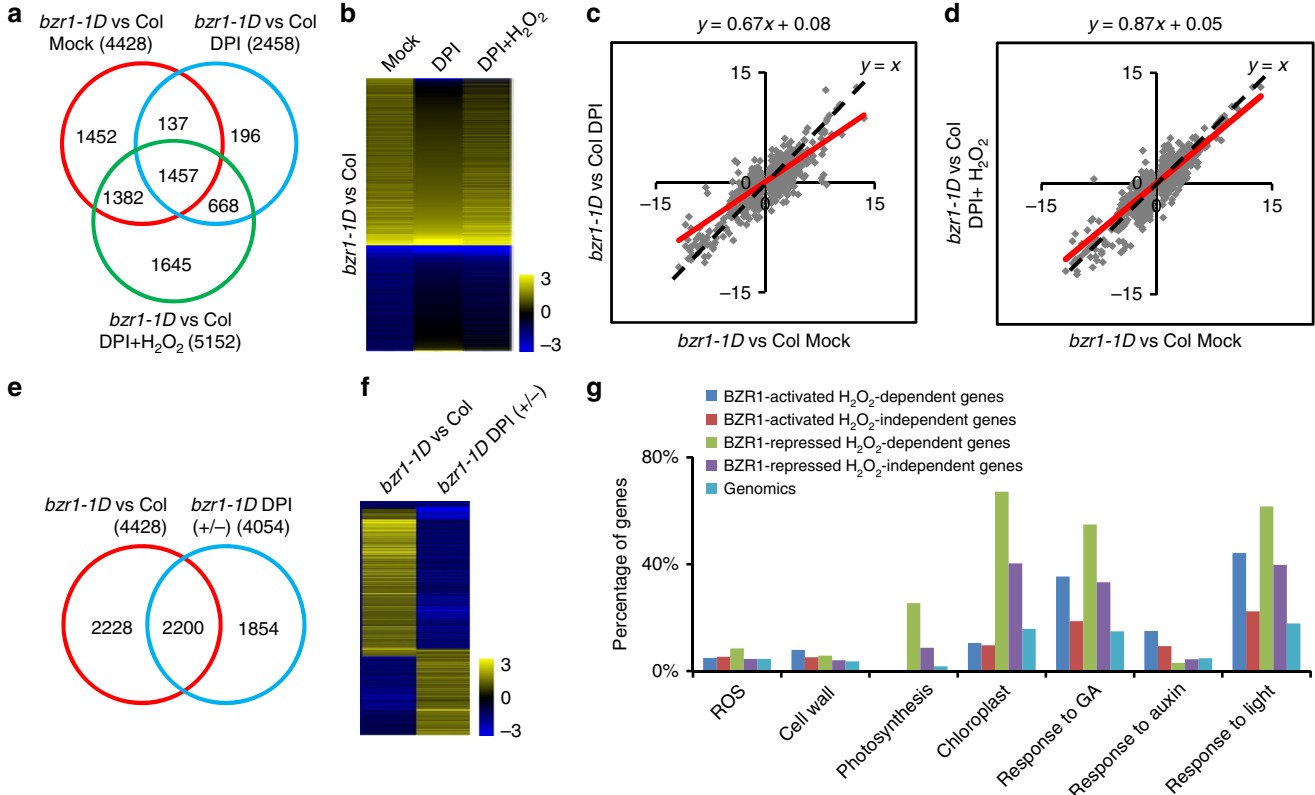

**Fig. 4** The accumulation of $H_2O_2$ is required for the transcriptional activity of BZR1. **a** The Venn diagram shows the overlap between sets of genes differentially expressed in *bzr1-1D* vs. Col-0 under different conditions: mock, medium containing 2 μM PPZ; DPI, medium containing 2 μM PPZ and 1 μM DPI; DPI + $H_2O_2$, medium containing 2 μM PPZ, 1 μM DPI, and 0.3 mM $H_2O_2$. The seedlings were grown in the dark for 6 days, and then gene expression was analyzed by RNA-Seq. **b** Hierarchical cluster analysis of the expression data of 4428 genes regulated by BZR1 under three different conditions. The numerical values for the green-to-red gradient bar represent log2-fold change values of *bzr1-1D* relative to Col-0. **c** Scatter plot of log2-fold change values of 4428 BZR1-regulated genes in the mock condition and in the DPI condition. The red line represents the trend line of the scatter plot. **d** Scatter plot of log2-fold change values of 4428 BZR1-regulated genes in the mock condition and in the DPI plus $H_2O_2$ condition. The red line represents the trend line of the scatter plot. **e** The Venn diagram shows the overlap between sets of genes regulated by BZR1 in the mock condition (*bzr1-1D* vs. Col-0), and regulated by DPI in the *bzr1-1D* background (*bzr1-1D* DPI+/−). **f** Hierarchical cluster analysis of the expression of genes co-regulated by BZR1 and DPI. **g** Gene ontology analysis of BZR1-regulated $H_2O_2$-dependent and $H_2O_2$-independent genes. Numbers indicate the percentages of genes belonging to each GO category

selected for phenotype analysis. Plants expressing *bzr1-1D* showed BR-activation phenotypes similar to the *bzr1-1D* mutant, with curled leaves and insensitivity to BR biosynthesis inhibitor PPZ (Fig. 3a–e). Transgenic plants expression *bzr1-1D^{C91,174,240S}* showed BR-enhanced phenotypes similar to *bzr1-1D*, whereas *bzr1-1D^{C63S}*, *bzr1-1D^{C63,73S}*, and *bzr1-1D^{5S}* transgenic plants all displayed normal leaf phenotypes and increased sensitivity to PPZ (Fig. 3a–e). The QC cell division ratio of *bzr1-1D^{5S}* (10%) was significantly lower than that of *bzr1-1D* transgenic plants (Fig. 3f). Moreover, conversion of the cysteine-84 of *bes1-D* to serine also suppressed the phenotypes of *bes1-D* (Supplementary Fig. 6c–f). Together, these results demonstrate that the conserved cys-63 of BZR1 and cys-84 of BES1 are essential for functions in various BR-mediated responses.

Next, we examined the effects of cysteine residues on BZR1 regulation of the expression of target genes. BZR1 is known to activate growth-related genes, such as *PRE1*, *EXP8*, and *IAA19*, to promote cell elongation. The expression levels of *PRE1*, *EXP8*, and *IAA19* were increased in *bzr1-1D-Ox* plants compared with that in wild-type plants, but were less increased in *bzr1-1D^{5S}-Ox* than in *bzr1-1D-Ox* (Fig. 3g). BZR1 is also known to repress BR biosynthesis genes, including *CPD* and *DWF4*, for feedback regulation of BR levels. Transcription levels of *CPD* and *DWF4* significantly decreased in *bzr1-1D-Ox* seedlings, while they were only slightly repressed in the *bzr1-1D^{5S}-Ox* seedlings, suggesting

that the mutations of cysteines impaired BZR1 activity (Fig. 3g). To more directly examine the functions of the oxidized residues in BZR1 regulation of gene expression, we performed transient gene expression assays using Arabidopsis mesophyll protoplasts. Co-expression of *bzr1-1D* significantly increased the expression of luciferase reporter driven by the *PRE5* promoter, whereas *bzr1-1D^{C63S}* only marginally increased *PRE5* promoter activity, and *bzr1-1D^{5S}* lost the transcriptional activation activity (Fig. 3h). These results indicate that mutagenesis of oxidized cysteine residues decreased the transcriptional activity of BZR1.

**$H_2O_2$ is required for the transcriptional activity of BZR1.** We carried out RNA-Seq analysis to determine whether $H_2O_2$ affects the BZR1-mediated gene expression in Arabidopsis genome. Wild-type and *bzr1-1D* mutant seedlings were grown in the dark for 6 days on the medium containing BR biosynthesis inhibitor PPZ, and/or NADPH oxidase inhibitor DPI, and/or $H_2O_2$. RNA-Seq analysis identified 4428 genes affected more than 1.5-fold by *bzr1-1D* comparing with wild-type in the presence of PPZ (BZR1-regulated genes in the mock condition). In the presence of PPZ and DPI, 2458 genes were affected by *bzr1-1D* (BZR1-regulated genes in the DPI condition), while 5152 genes were affected by *bzr1-1D* in the presence of PPZ, DPI, and $H_2O_2$ (BZR1-regulated genes in the DPI plus $H_2O_2$ condition) (Fig. 4a and Supplementary Data 1–3). Among the 4428 genes regulated by BZR1 in

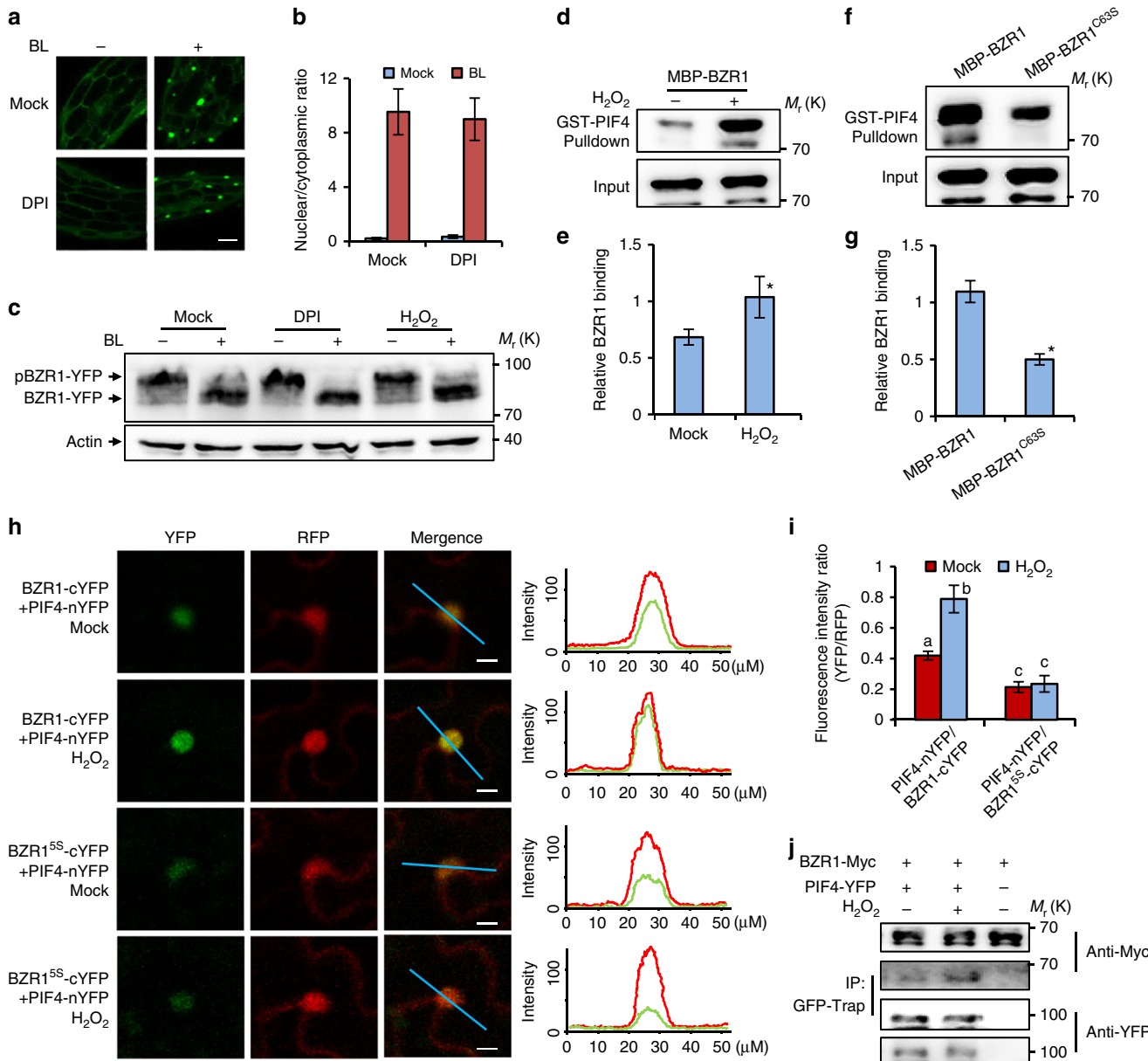

**Fig. 5** $H_2O_2$ enhances the interaction between BZR1 and PIF4 in vitro and in vivo. **a** DPI treatment had no significant effects on BL-induced nuclear localization of BZR1-YFP. Transgenic Arabidopsis plants expressing *BZR1-YFP* were grown on ½ MS medium containing 2 μM PPZ or 2 μM PPZ plus 1 μM DPI in the dark for 5 days, and then treated with 100 nM BL or mock solution for 3 h. Scale bar, 20 μm. **b** Quantification of fluorescent intensities ratio between nuclear and cytoplasmic signals in **a**. The standard errors calculated from 50 cells for each treatment. **c** Immunoblot analysis of phosphorylated and dephosphorylated BZR1 in 5-day-old dark-grown *pBZR1:BZR1-YFP* using anti-YFP antibody. Actin bands showed protein loadings. **d**–**g** In vitro pull-down assays showed that $H_2O_2$ enhanced, but Cys-63 mutation to Ser reduced, the BZR1 binding to PIF4. **e**, **g** show the quantification of pull-down assays (normalized to input) in the **d** and **f**, respectively. Error bars represented the s.d. of three independent experiments. *$p < 0.05$, as determined by a Student's *t*-test. **h**, **i** rBiFC confocal images showed that $H_2O_2$ enhanced, but mutation of cysteines reduced the interaction between BZR1 and PIF4 in plants. The fluorescent signals of YFP (BiFC) and RFP (reference) were determined along a line drawn on the confocal images using ImageJ software. Error bars, s.d. (*n* = 50 images). Different letters above the bars indicated statistically significant differences between the samples (two-way ANOVA, $p < 0.05$). Scale bar, 10 μm. **j** Co-immunoprecipitation (CoIP) assays showed that $H_2O_2$ increased the interaction between BZR1 and PIF4 in plants. The Arabidopsis mesophyll protoplast transformed with BZR1-Myc and/or PIF4-YFP were treated with mock solution or 100 μM $H_2O_2$ for 1 h, then immunoprecipitation was performed using GFP-Trap agarose beads, and the immunoblots were probed using anti-Myc and anti-YFP antibodies

the mock condition, the expression of 2834 genes (64%) was no longer responsive to BZR1 in the DPI-treated $H_2O_2$-deficient condition, while 1382 genes (48%) out of the 2834 genes recovered responsiveness to BZR1 when $H_2O_2$ was added back to the $H_2O_2$-deficient condition (Fig. 4a, b). Scatter plot analysis showed that the slope of the trend line in the plot comparing BZR1-regulated genes in the mock condition with that in the DPI

condition was only 0.67, but was restored to 0.87 in the plot comparing the BZR1 response genes in the mock condition with that in the DPI plus $H_2O_2$ condition (Fig. 4c, d). The overall effect of BZR1 on gene expression was significantly reduced by DPI treatment, but this reduction was effectively reversed by $H_2O_2$ co-treatment, suggesting $H_2O_2$ plays a positive role in BZR1-regulated gene expression.

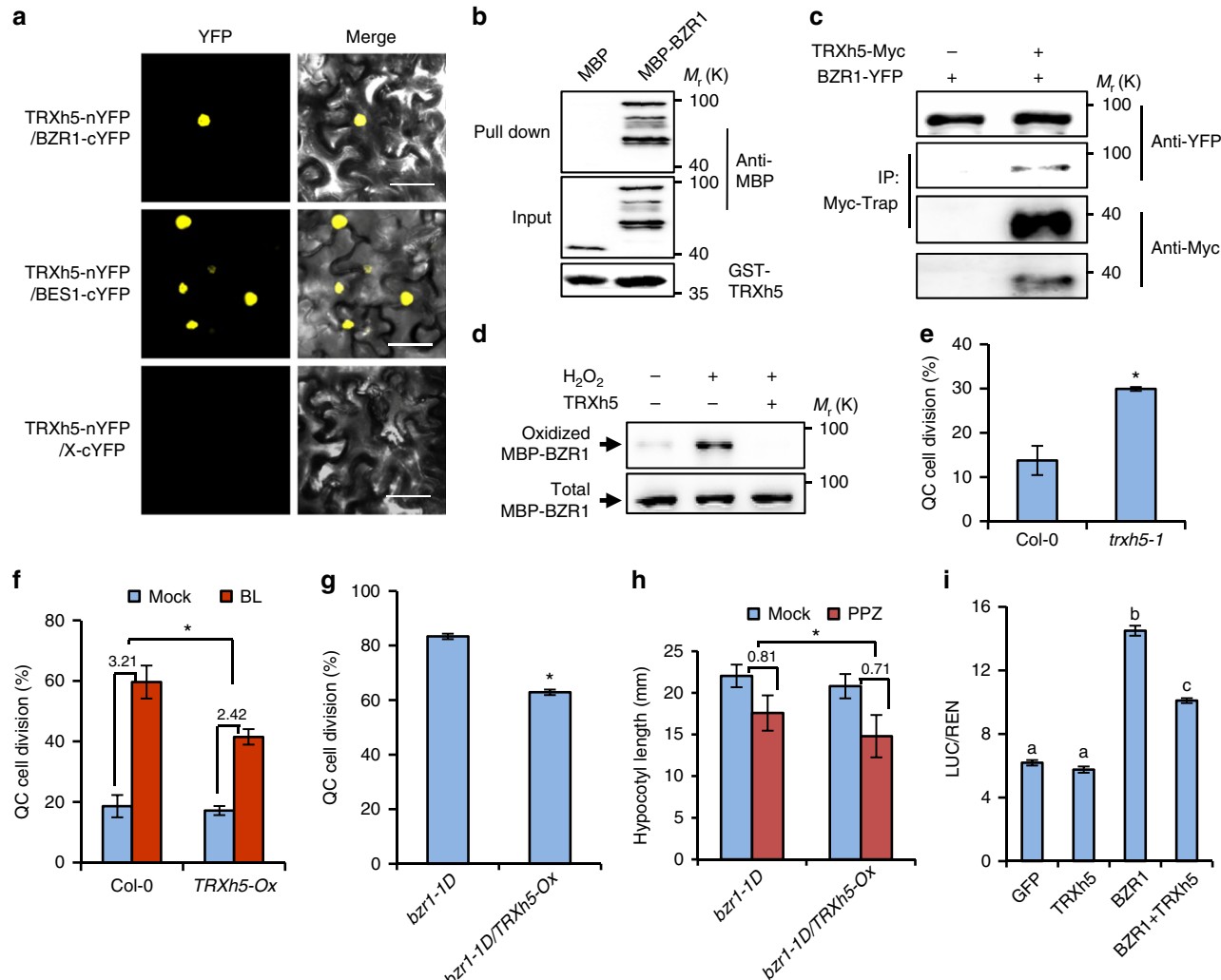

**Fig. 6** TRXh5 is a redox mediator of BZR1. **a** BiFC assays showed TRXh5 interaction with BZR1 and BES1 in tobacco leaf cells. Scale bar, 50 μm. **b** TRXh5 directly interacted with BZR1 in vitro. Glutathione agarose beads containing GST-TRXh5 were incubated with equal amount of MBP or MBP-BZR1. Proteins bound to GST-TRXh5 were detected by immunoblot with anti-MBP antibody. **c** TRXh5 interacts with BZR1 in plants. Immunoprecipitation was performed using Myc-trap beads and transgenic Arabidopsis plants expressing *pBZR1:BZR1-YFP* only or co-expressing *pBZR1:BZR1-YFP* and *p35S:TRXh5-Myc*, and the immunoblot analyzed using anti-Myc or anti-YFP antibodies. **d** TRXh5 catalyzed reduction of $H_2O_2$-oxidized BZR1 in vitro. MBP-BZR1 pretreated with $H_2O_2$ was incubated with or without a TRX system consisting of TRXh5, NTRA, and NADPH for 3 h at room temperature, and then analyzed by biotin-switch methods. **e**–**g** Quantification of QC cell division in the root of *trxh5-1* mutant (**e**), *TRXh5-Ox* (**f**), *bzr1-1D* and *bzr1-1D/TRXh5-Ox* (**g**) with or without BL treatment as indicated. At least 50 seedlings were examined for each biological repeat. Error bars represented the s.d. of three independent experiments. *$p < 0.05$, as determined by a Student's *t*-test. **h** *TRXh5* overexpression attenuated the resistance of *bzr1-1D* to PPZ. The *bzr1-1D* and *bzr1-1D/TRXh5-Ox* seedlings were grown on ½ MS medium containing mock solution or 2 μM PPZ in the dark for 6 days. Error bars indicated s.d. ($n = 30$ plants). *$p < 0.05$, as determined by a Student's *t*-test. **i** TRXh5 reduced the transcriptional activity of BZR1 in protoplast transient assay of the *pPRE5:LUC* reporter gene. Error bars, s.d. ($n = 3$). Different letters above the bars indicated statistically significant differences between the samples (two-way ANOVA, $p < 0.05$)

To further define $H_2O_2$-dependent BZR1-regulated genes, we analyzed the genes differentially expressed in *bzr1-1D* mutant with or without DPI treatment, and identified 4054 genes affected >1.5-fold by DPI (Fig. 4e and Supplementary Data 4). Among the 4428 genes affected by *bzr1-1D* in comparison with wild-type, 2200 genes (49.7%) were also affected by DPI treatment, which we named $H_2O_2$-dependent BZR1-regulated genes, and the other 2228 genes were named $H_2O_2$-independent BZR1-regulated genes (Fig. 4e). Among the 2200 $H_2O_2$-dependent BZR1-regulated genes, 2053 genes (93.3%) were affected in the opposite way by DPI treatment and *bzr1-1D* vs. wild-type, confirming that DPI inhibits the transcriptional activity of BZR1 (Fig. 4f).

Gene ontology (GO) analysis showed that the $H_2O_2$-dependent BZR1-regulated genes play dominant roles in BZR1-mediated biological processes and cellular activities (Fig. 4g). For example, genes involved in cell growth and auxin responses were highly enriched in $H_2O_2$-dependent BZR1-activated genes. In contrast, the genes involved in ROS production, photosynthesis, and chloroplasts were highly enriched in $H_2O_2$-dependent BZR1-repressed genes. Genes involved in gibberellic acid (GA) and light responses were highly enriched in both $H_2O_2$-dependent BZR1-activated and BZR1-repressed genes (Fig. 4g). Previous studies showed that BZR1 interacts with PIF4 and ARF6 to form BZR1-PIF-ARF module to regulate cell elongation downstream of multiple hormonal and environmental signals[31,32]. GO analysis showed that PIF-induced genes and IAA3-repressed genes were highly enriched in the $H_2O_2$-dependent BZR1-activated genes, while PIF-repressed genes were highly enriched in the $H_2O_2$-dependent BZR1-repressed genes, suggesting $H_2O_2$ plays

important roles in the regulation of BZR-PIF-ARF common target gene expression (Supplementary Fig. 7a, b).

**Oxidation promotes BZR1 binding with PIF4 and ARF6.** To learn how $H_2O_2$ regulates the activity of BZR1, we examined whether $H_2O_2$ affects the subcellular localization, protein levels, and phosphorylation status of BZR1 by analyzing BZR1-YFP proteins in the *pBZR1:BZR1-YFP* transgenic plants. The results showed that BR treatment induced the transport of BZR1 from the cytoplasm to the nucleus regardless of DPI presence, suggesting that reducing the content of $H_2O_2$ does not affect BR-regulated nucleo-cytoplasmic shuttling of BZR1 (Fig. 5a, b). Treatment with BR also caused dephosphorylation of BZR1, but co-treatment with $H_2O_2$ or DPI had no obvious effects on the levels of phosphorylated and unphosphorylated BZR1 (Fig. 5c). These results indicated that disruption of $H_2O_2$ homeostasis does not have a significant effect on BR levels or BR signaling upstream of BZR1.

To determine whether $H_2O_2$ affects the DNA-binding ability of BZR1, we performed DNA-protein pull-down and chromatin immunoprecipitation quantitative PCR (ChIP-qPCR) assays. The results showed that BZR1 specifically binds to the promoter of *SAUR15*, which is a known BZR1-binding target gene. $H_2O_2$ treatment did not enhance in vitro binding activity of BZR1 to the promoter of *SAUR15* in the absence or presence of PIF4 and ARF6 (Supplementary Fig. 8a–c). Similarly, ChIP-qPCR analysis showed that $H_2O_2$ did not have a significant effect on BZR1 DNA-binding capacity in plants (Supplementary Fig. 8d), suggesting that $H_2O_2$ does not affect DNA-binding ability of BZR1. In contrast, $H_2O_2$ significantly improved the transcriptional activity of BZR1, PIF4, and ARF6 in protoplast transient assays (Supplementary Fig. 8e).

The enrichment of PIF and ARF-bound/regulated genes among the $H_2O_2$-dependent BZR1-regulated genes (Supplementary Fig. 7a, b) suggests that $H_2O_2$ may affect the interactions between BZR1 and PIF4 or ARF6. To test this hypothesis, we analyzed the effects of $H_2O_2$ on the binding affinity of BZR1 to PIF4 and ARF6. The in vitro pull-down assays showed that glutathione-S-transferase (GST)-PIF4 interacted with MBP-BZR1, while $H_2O_2$ treatment dramatically increased the interaction between BZR1 and PIF4 (Fig. 5d, e). Mutagenesis of cysteine to serine residues reduced the binding ability of BZR1 to PIF4 (Fig. 5f, g). Using GST-ARF6, we observed that $H_2O_2$ increased the interaction between BZR1 and ARF6 (Supplementary Fig. 9a–d). These results suggested that $H_2O_2$-induced oxidation enhances BZR1-binding affinity with its partner, PIF4 and ARF6 in vitro.

To verify the $H_2O_2$ effects on the interactions of BZR1 with PIF4 and ARF6 in plants, we analyzed these interactions using a newly developed ratiometric bimolecular fluorescence complementation (rBiFC) assay[33]. In this assay, BZR1 or BZR1[5S] and PIF4 or ARF6 were simultaneously cloned into a single vector backbone containing a monomeric red fluorescent protein driven by the *35S* promoter as an internal marker for expression control and ratiometric analysis. Consistent with previous results, BZR1 interacted with PIF4 and ARF6 in the epidermal cells of tobacco leaves, but this interaction was significantly enhanced by $H_2O_2$ treatment (Fig. 5h, i and Supplementary Fig. 9e, f). However, co-expression of BZR1[5S] and PIF4 or ARF6 resulted in very weak YFP signals, and $H_2O_2$ treatment had no significant effect on the interactions between BZR1[5S] and PIF4 or ARF6 (Fig. 5h, i and Supplementary Fig. 9e, f). We further tested the effect of $H_2O_2$ on the interactions of BZR1 with its partner PIF4 and ARF6 in plants using co-immunoprecipitation assays. The results showed that $H_2O_2$ clearly enhanced the interactions between BZR1 and PIF4 or ARF6 in plants (Fig. 5j and Supplementary Fig. 9g). Together,

these results demonstrated that $H_2O_2$-induced oxidation of BZR1 promotes its interactions with PIF4 and ARF6.

**TRX-h5 is a redox mediator of BZR1.** To counter the effects of oxidation-facilitated BZR1 activation, reducing agents must be engaged to catalyze the switch of BZR1 from oxidized to reduced form. In mammals, the cytosolic/nuclear thioredoxin TRX1 has been well characterized as a master transcriptional regulator through direct or indirect interactions with different transcription factors[3]. The plant *h*-type thioredoxin members have been reported to localized in cytoplasm and nucleus, and behave similarly to the mammalian TRX1 system[34]. To test whether members of TRX-h family interact with and mediate the reduction of BZR1, we performed the BiFC assay in tobacco leaves by co-expressing BZR1 or BES1 fused to the carboxyl-terminal half of yellow fluorescent protein (BZR1-cYFP or BES1-cYFP) and members of TRX-h proteins fused to the amino-terminal half of YFP (nYFP). Strong YFP fluorescent signals were observed in the nucleus when TRXh5-nYFP was co-transformed with BZR1-cYFP or BES1-cYFP, indicating a specific TRXh5 interaction with BZR1 and BES1 (Fig. 6a and Supplementary Fig. 10). In vitro pull-down assays showed that GST-TRXh5 directly interacted with MBP-BZR1, but not with MBP alone (Fig. 6b). Gene expression analysis using the Arabidopsis microarray data displayed in the eFP browser indicated very similar ubiquitous expression patterns of *TRXh5* and *BZR1* (Supplementary Fig. 11a, b), suggesting they may function together to regulate plant growth and development. Consistent with the BiFC and in vitro assays, co-immunoprecipitation assays showed that TRXh5 interacts with BZR1 in plants (Fig. 6c). To examine whether TRXh5 can catalyze the reduction of BZR1, we carried out the in vitro reduction reaction. Recombinant BZR1 proteins were first oxidized by $H_2O_2$ treatment, and then incubated with recombinant TRXh5 and NTR isoform A (NTRA). The result showed that TRXh5 caused a decrease of the level of oxidized BZR1, suggesting TRXh5 catalyzes BZR1 reduction (Fig. 6d).

To determine whether the thioredoxin system regulates the function of BZR1 in plants by regulating its oxidized state, we characterized the phenotype of TRXh5 knock out mutant (*trxh5-1*), and transgenic plants overexpressing *TRXh5* (*TRXh5-Ox*). Microscopic observation showed that *trxh5-1* mutant exhibited higher frequency of QC cell division than wild-type plants (Fig. 6e). On the other hand, overexpression of *TRXh5* partially suppressed the BR-induced and BZR1-induced QC cell division (Fig. 6f, g). *TRXh5* overexpression not only partly restored the sensitivity of *bzr1-1D* to PPZ in the dark (Fig. 6h and Supplementary Fig. 12a, b), but also attenuated the large-leaf phenotype of *bzr1-1D* under light (Supplementary Fig. 12c–g). Furthermore, in transient assays TRXh5 reduced the transcription activity of BZR1 (Fig. 6i). Together, these results demonstrate that TRXh5 negatively regulates BR signaling by changing the oxidation state of BZR1.

## Discussion

Among ROS, $H_2O_2$ in particular plays indispensable roles in a wide range of plant growth and developmental processes, as well as in stress responses. $H_2O_2$ is referred to as a second messenger integrating a large number of signals to orchestrate correct cellular responses. However, the molecular targets and underlying mechanism of regulation have remained unclear. In this study, we demonstrated that BR increases cellular level of $H_2O_2$, which causes oxidation of BZR1 at a conserved cysteine residue; this oxidation enhances BZR1 transcriptional activity by promoting its interaction with ARF and PIF4. The oxidized BZR1 is

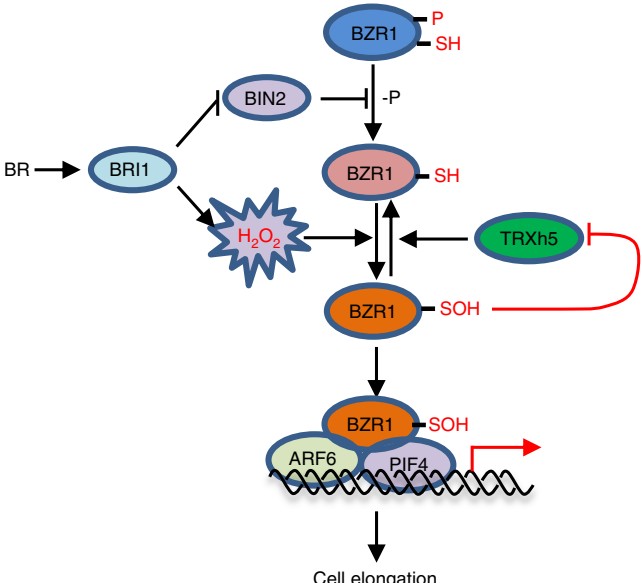

**Fig. 7** A model for the regulatory network integrating BR and $H_2O_2$ signals in cell elongation. Binding of BR to receptor kinase BRI1 not only inhibits the kinase activity of BIN2 to promote dephosphorylation of BZR1, but also increases cellular level of $H_2O_2$, which causes oxidation of BZR1 at a conserved cysteine residue. This oxidation enhances BZR1 transcriptional activity by promoting its interaction with partners, such as ARF6 and PIF4. The oxidized BZR1 is reduced by thioredoxin TRXh5. As the level of $H_2O_2$ and TRXh5 activity can be altered by many signals, such redox regulation of BZR1 plays an important role in fine-tuning BR responses by additional signals

reduced by thioredoxin TRXh5. Such redox regulation of BZR1 plays an important role in BR signal transduction (Fig. 7).

Several studies have reported that BR-induced $H_2O_2$ accumulation is necessary for numerous BR-mediated biological processes, including stomatal movement[15,16], salt tolerance[35], responses to heat and oxidative stresses[36,37]. In this study, we showed that BR-induced $H_2O_2$ production is also critical for BR promotion of cell elongation and root QC cell division. Chemical scavenging of $H_2O_2$, silencing *RBOHD* and *RBOHF* expression, and overexpression of catalase genes, all compromised BR- and/ or activated BZR1-induced cell elongation and QC cell division. DPI treatment decreased BR sensitivity in a dose-dependent manner in the hypocotyl elongation assays. These results indicated that $H_2O_2$ is essential for diverse BR-mediated plant biological processes, and $H_2O_2$ plays an important role in BR signal transduction.

Multiple components of the BR signaling pathway have been reported to be the modification targets of ROS[24,38]. BAK1 was shown to be glutathionylated by glutaredoxin AtGRXC2 via a thiol-dependent reaction with GSSG, which inhibits the kinase activity[38]. Nitric oxide inhibits the kinase activity of BIN2 by S-nitrosylation of cys-162, which is conserved in GSK3 kinase family[24]. However, these studies only provided the in vitro data, and whether redox regulates the kinase activity of BAK1 and BIN2 in plants still remains unknown.

In the present study, we demonstrate, through several lines of evidence, that $H_2O_2$ is involved in BR signal transduction through oxidation of the master transcription factor BZR1. First, chemical scavenging of $H_2O_2$ suppressed the long hypocotyl and high frequency of QC cell division in *bin2bil1bil2* mutants and wild-type plants treated with GSK3 inhibitors bikinin or LiCl. Further, the *BZR1ΔDM* transgenic plants, which express a mutant BZR1 protein that cannot be phosphorylated by BIN2, exhibited DPI

inhibition of cell elongation and QC cell division, suggesting that DPI inhibits BR responses independent of BIN2/GSK3s. Second, $H_2O_2$ and its biosynthesis inhibitor DPI had no significant effect on the protein level or on the phosphorylation status of BZR1, confirming that $H_2O_2$ does not affect BR signaling upstream of BZR1. Third, BIAM labeling assays showed that $H_2O_2$ effectively induced the oxidation of BZR1 in vivo and in vitro. Finally, mutation of the $H_2O_2$-oxidized cysteine residues abolished in vitro $H_2O_2$ oxidation of BZR1 and abrogated BZR1's activities in activating gene expression and promoting cell elongation and QC cell division. Together, these results demonstrated that $H_2O_2$ contributes to BR signal transduction downstream of GSK3 kinase and through oxidation of the transcription factor BZR1.

Transcriptomic analyses further support that the transcriptional activity of BZR1 is under redox regulation and that BZR1 is a major factor mediating transcriptional responses to $H_2O_2$. Under an $H_2O_2$-deficient condition, the amplitude of BZR1-regulated gene expression was significantly reduced and approximately 64% of the genes were no longer altered in *bzr1-1D*, while adding $H_2O_2$ back to the $H_2O_2$-deficient condition partially restored the responsiveness to *bzr1-1D* for about 58% of the BZR1-regulated genes. On the other hand, 2200 (54%) of the 4054 DPI-altered genes were also altered by *bzr1-1D*, with 93.3% of them affected oppositely by DPI and *bzr1-1D*. Such strong correlation between DPI and BZR1 effects of transcriptome supports that BZR1 is a major factor for $H_2O_2$ modulation of gene expression under our experimental conditions.

GO comparison of BZR1-regulated genes under oxidized and reduced states suggests that the redox regulates BZR1 to modulate many cellular functions. $H_2O_2$-dependent BZR1-repression genes are enriched in photosynthesis and ROS production. BZR1 repressed the expression of several members of thioredoxins, glutaredoxins, and peroxidase family genes, including *TRXf1*, *TRXm1*, *TRXm2*, *TRXm4*, *TRXy2*, *TRXz*, *GRX2*, and *PRXQ* via an $H_2O_2$-dependent manner (Supplementary Data 1). Such BZR1 repression of ROS-related genes may contribute to feedback regulation and redox homeostasis. $H_2O_2$ enhances BZR1 activation of genes involved in cell wall biogenesis and responsive to GA, auxin, and light, consistent with its function in promoting BR-induced cell elongation. The $H_2O_2$-dependent BZR1-regulated genes are enriched with PIF4-regulated or IAA3-regulated genes. Such significant overlaps with auxin and light responses are consistent with the recent finding that auxin, light, and BR regulate shoot cell elongation through a central module of BZR1 interaction with the PIF4 and ARF6 factors[31]. Indeed, we found that $H_2O_2$ oxidation of BZR1 enhances its interaction with PIF4 and ARF6. As such, BR and $H_2O_2$ control BZR1 activity through distinct mechanisms of enhancing nuclear localization/ DNA binding and interaction with partners, respectively. Such a specific mechanism of $H_2O_2$ regulation is likely to allow redox regulation of subsets of BR-regulated processes.

$H_2O_2$-mediated oxidative modification is a reversible process, and in many cases the reduction of oxidized cysteine residues is carried out by the thioredoxin system[34,39]. Thioredoxins are typically small proteins that are conserved in all free-living organisms and are able to catalyze the reduction of the disulfide bond in their substrates and the denitrosylation of potential protein-SNO modifications[34]. Detailed characterization of TRXh5 in this study reveals that the growth-promoting hormone BR represses the expression of *TRXh5* through BZR1 (Supplementary Fig. 13a, b) and that TRXh5 negatively regulates BR signal transduction by catalyzing BZR1 oxidation-to-reduction switch. Previous studies have shown that TRXh5 regulates SA-dependent plant immunity by selective protein denitrosylation[25,40]. Pathogens and SA highly upregulate the expression of *TRXh5*, which reduces NPR1 to cause its oligomer-monomer

exchange and nuclear localization to activate immune responses. As such, TRXh5 is activated by pathogens but repressed by BR, and it activates immunity and represses growth, and thereby functions as a major node of crosstalk that mediates a trade-off between growth and immunity in plants.

The proper balance among stem cell maintenance, proliferation, and differentiation is a key aspect of development in multicellular organisms, and is controlled by gradient signals such as BR and ROS[41,42]. BR and activated BZR1 are patterned in the Arabidopsis root tip, with low levels of nuclear-localized BR-activated BZR1 in the stem cell niche to maintain stem cell and meristem size, and high levels of nuclear BZR1 in the epidermal cells in the transition-elongation zone to promote cell elongation[41]. $H_2O_2$ also accumulates mainly in the expanding cells of the elongation zone to promote cellular differentiation, whereas superoxide mainly accumulates in the dividing cells of the meristem zone to maintain stem cell[42]. BR treatment not only induces the nuclear localization of BZR1 and the accumulation of $H_2O_2$, but also reduces the content of superoxide, as the BR-deficient mutant det2-9 was recently shown to over accumulate superoxide in the meristem zone, which leads to the inhibition of root growth[43]. Overall, the distribution pattern of $H_2O_2$ is similar to that of nuclear BZR1 (Supplementary Fig. 14a, b), consistent with $H_2O_2$ modulating BZR1 activity in root growth regulation. The gradient pattern of $H_2O_2$ depends on the basic/helix-loop-helix transcription factor UPBEAT1 (UPB1), which regulates the expression of peroxidases and root growth rate[42]. While the role of $H_2O_2$ in QC function remains unclear. BZR1 has been shown to promote QC division in response to DNA damage stress[20]. The $H_2O_2$ enhancement of the transcription activity of BZR1 could potentially provide dual regulation of root growth by BR and redox signals, important for optimal growth according to endogenous and environmental signals.

The redox regulation of BZR1 by $H_2O_2$ and TRXh5 establishes a molecular framework of $H_2O_2$ crosstalk with BR signaling. As, $H_2O_2$ is involved in numerous plant physiological processes and stress responses, this mechanism of crosstalk is likely important for coordinating plant growth with environmental adaptation.

## Methods

**Plant materials and growth condition.** *Arabidopsis thaliana* plants were grown in a greenhouse or a growth chamber with a 16-h light/8-h dark cycle at 22–24 °C for general growth and seed harvesting. All wild-type, various mutants, and transgenic plants in this study are in Col-0 ecotype background, except bin2bil1bil2 mutant in the Ws background. These include *bzr1-1D, bri1-116, trxh5-1, rbohDrbohF, p35S: CAT2, p35S:CAT3, p35S:TRXh5-YFP, p35S:TRXh5-YFP/bzr1-1D, p35S:BZR1-YFP, p35S:BZR1ADM-YFP, p35S:YFP, p35S:bzr1-1D-YFP, p35S:bzr1-1D^{C63S}-YFP, p35S: bzr1-1D^{C6,73S}-YFP, p35S:bzr1-1D^{C91,137,240S}-YFP, p35S:bzr1-1D^{C63,73,91,174,240S}- YFP, p35S:bes1-D-YFP, p35S:bes1-D^{C84S}-YFP,* and *p35S:bes1-D^{C84, 94, 261S}-YFP*. For hypocotyl length measurement, seedlings were photocopied and their hypocotyl lengths were measured using ImageJ software (http://rsb.info.nih.gov/ij).

**Vector construction.** Full length cDNA of *TRXh5, BZR1, bzr1-1D, BES1,* and *bes1-D* without stop codon were amplified by PCR and cloned into pENTR™/SD/D-TOPO™ vectors (ThermoFisher), and then recombined with destination vector pX-YFP (35S:C-YFP), pX-nYFP (35S:C-nYFP), pX-cYFP (35S:C-cYFP), pDEST15 (N-GST), and pMAL2CGW (N-MBP). All mutants of *BZR1, bzr1-1D,* and *bes1-D* were generated using Quick-change Site-directed Mutagenesis kit (Stratagene), cloned into pENTR™/SD/D-TOPO™ vectors, and then recombined into destination vector pX-YFP (35S:C-YFP). Oligo primers used for cloning are listed in Supplemental Data 5.

**Microscopy analysis.** Phenotypic analysis of roots were carried out with modified Pseudo-Schiff Propidium Iodide (mPS-PI) staining method[44]. Seedlings of wild-type and various mutants for analysis of QC cell division were grown on ½ MS basal salt medium supplemented with 1% sucrose for 4 days in the continues light, and then transferred to ½ MS liquid medium containing different hormones and chemicals as indicated for another 1 day under light conditions. Whole seedlings were fixed in fixative buffer (50% methanol and 10% acetic acid) and stained using the mPS-PI staining method. The cellular organization of root tips was analyzed from longitudinal optical sections obtained using an LSM700 laser scanning

confocal microscope (Zeiss). At least 50 roots were analyzed for each treatment in each experiment and each experiment was repeated independently at least three times. The statistical significance of differences between means was determined using the Student's *t*-test.

**BIAM labeling assay.** MBP and MBP-BZR1 proteins were purified from *E. coli* using amylose resin (NEB), and then treated with different concentrations of $H_2O_2$ at room temperature for 15 min. The proteins were precipitated by adding one volume of acetone at −20 °C for 20 min and centrifuged at $5000 \times g$ for 5 min. The pellets were washed three times with 50% acetone and dissolved in 500 µl labeling buffer (50 mM MES-NaOH, pH 6.5, 100 mM NaCl, 1% TritonX-100, 100 µM BIAM), and then incubated at room temperature in the dark for 1 h. The labeling reactions were terminated by the addition of β-mercaptoethanol to a final concentration of 20 mM. The reaction mixtures were precipitated by adding one volume of acetone at −20 °C for 20 min and centrifuged at $5000 \times g$ for 5 min. The pellets were dissolved in 50 µl SDS sample buffer, and subjected to separate on SDS-PAGE. Proteins labeled with BIAM were detected with HRP-conjugated streptavidin[30] (Cell Signaling, Cat: 7075S, 1:5000 dilution). An antibody against MBP was used to show the total MBP or MBP-BZR1 proteins (NEB, Cat: 8038L, 1:5000 dilution).

**In vivo and in vitro biotin-switch assays.** The slightly modified biotin switch assays[30,45] that selectively add biotin to reduced or oxidized thiol groups were used to analyze the redox status of BZR1 proteins in Arabidopsis plants or purified recombinant BZR1 proteins treated with $H_2O_2$.

For in vivo oxidation analysis, the *p35S:BZR1-YFP* transgenic seedlings were grown in ½ MS liquid medium containing 1% sucrose for 7 days under 16-h light/ 8-h dark condition, and then treated with or without 100 nM BL, and/or 1 mM $H_2O_2$ for another 3 h under light conditions. Seedlings were harvested and ground to fine powder in liquid nitrogen. Proteins were extracted in EBR buffer (20 mM HEPES, pH 8.0, 40 mM KCl, 5 mM EDTA, 0.5% TritonX-100, 1% SDS, 1 mM PMSF, and 1× protease inhibitor cocktail).

For in vitro oxidation analysis, the maltose-binding protein-BZR1 fusion (MBP-BZR1) protein was purified from *E. coli*, and then treated with 1 mM $H_2O_2$ or 1 mM DTT at room temperature for 15 min. The proteins were precipitated by adding one volume of acetone at −20 °C for 20 min and centrifuged at $5000 \times g$ for 5 min. The pellet was washed three times with 50% acetone and dissolved in 500 µl EBR buffer.

To detect the oxidized form of BZR1, the plant extracts or the recombinant BZR1 proteins were incubated with 100 mM NEM in the EBR buffer at room temperature for 30 min with frequent vortexing to block free thiol. The samples were precipitated with one volume of acetone and washed three times with 50% acetone. The pellets were dissolved in 500 µl EBR buffer plus 20 mM DTT, and then incubated at 37 °C for 30 min to reduce the oxidized thiols. DTT was removed by protein precipitation and pellets were resuspended in 500 µl EBR buffer. The supernatant was labeled with 100 µM BIAM at room temperature for 1 h in the dark, and proteins were precipitated with one volume of acetone to remove free BIAM.

The BIAM-treated proteins were then dissolved in 250 µl EBR buffer and diluted by 750 µl NEB buffer (20 mM HEPES, pH 8.0, 40 mM KCl, 5 mM EDTA, 0.25% TritonX-100, 1 mM, PMSF and 1× protease inhibitor cocktail). After centrifugation at $10,000 \times g$ for 5 min, the supernatant was added to 40 µl streptavidin beads (Biomag) and incubated at 4 °C for overnight. Beads were washed five times with NEB buffer, and proteins were eluted by 50 µl 2× SDS sample buffer and the samples were separated on 8% SDS-PAGE gels. An aliquot of proteins before incubation with streptavidin beads were also analyzed as total BZR1 protein. The gel blots were probed with anti-GFP antibody (homemade, 1:5000 dilution) or anti-MBP antibody (NEB, Cat: E8038L, 1:5000 dilution). The uncropped gel/blot images were presented in the Supplementary Fig. 15.

**Mass spectrometric analysis of oxidized residues in BZR1.** Briefly, purified MBP-BZR1 protein labeled with biotin-IAM was digested in-gel by trypsin, and then analyzed by liquid chromatograph tandem mass spectrometry (LC-MS/MS) using a ThermoFisher Orbitrap Fusion mass spectrometer. MS1 scans were acquired in the Orbitrap MS using a mass resolution of 240,000, and MS2 scans were obtained from Orbitrap MS with a resolution of 30,000. The raw data from mass spectrometry was searched against the NCBI Arabidopsis protein database using the Protein Prospector searching software (http://prospector.ucsf.edu/ prospector/mshome.htm), with a 10-ppm tolerance of the precursor and a 20-ppm tolerance for higher energy collision conditions (HCD) MS/MS. Cysteine biotinylation (C14H22N4O3S, mass change = 326.14) were included in the search.

**Quantitative reverse transcriptase-PCR analysis.** Total RNA was extracted from 6-day-old Arabidopsis seedlings of wild-type, *bzr1-1D*, and transgenic plants using the Trizaol RNA extraction kit (Transgene). The first-strand cDNAs were synthesized using RevertAid reverse transcriptase (Thermo) and used as RT-PCR templates. Quantitative PCR analyses were performed on a CFX connect real-time PCR detection system (Bio-Rad) using a SYBR green reagent (Roche) with gene-specific primers (see Supplementary Data 5).

**In vitro pull-down assays**. The recombinant GST-fused PIF4, ARF6, or TRXh5 were purified from bacteria using glutathione beads (GE Healthcare). BZR1 or BZR1$^{C63S}$ fused to MBP were purified using amylose resin (NEB). Glutathione beads containing 1 µg of GST-PIF4, GST-ARF6, or GST-TRXh5 were incubated with 1 µg MBP, MBP-BZR1, or MBP-BZR1$^{C63S}$ as indicated in pull-down buffer (20 mM Tris-HCl, pH 7.5, 100 mM NaCl, 1 mM EDTA) with or without 1 mM $H_2O_2$ at 4 ℃ for 1 h, and the beads were washed five times with wash buffer (20 mM Tris-HCl, pH 7.5, 300 mM NaCl, 0.1% Nonidet P-40, 1 mM EDTA). The proteins were eluted from beads by boiling in 50 µl 2× SDS sample buffer and separated on 8% SDS-PAGE gels. Gel blots were analyzed using anti-MBP (NEB, Cat: E8038L, 1:5000 dilution) and anti-GST antibodies (Merck, Cat: 16–209, 1:3000 dilution).

**Co-immunoprecipitation assays**. Plants or mesophyll protoplast expressing different constructs as indicated were extracted with NEB buffer (20 mM HEPES-KOH, at pH 7.5, 40 mM KCl, 1 mM EDTA, 0.5% Triton X-100, and 1× protease inhibitors, Roche). After centrifugation at $20,000 \times g$ for 10 min, the supernatant was incubated with GFP-Trap agarose beads (Chromotek) for 1 h, and the beads were washed four times with wash buffer (20 mM HEPES-KOH, at pH 7.5, 40 mM KCl, 1 mM EDTA, 0.05% Triton X-100, and 0.1% NP40). The proteins were eluted from the beads by boiling with 2× SDS sample buffer, and analyzed by SDS-PAGE and immunoblotted with anti-YFP (homemade, 1:5000 dilution) and anti-Myc (Sigma, Cat: M4439, 1:5000 dilution) antibodies.

**rBiFC assays**. Full length of *BZR1* or *BZR1*$^{5S}$ were amplified by PCR and cloned into the pDONR221-P1P4 vector, *PIF4* or *ARF6* were cloned into the pDONR221-P3P2 vector using the BP recombination reaction (Invitrogen), respectively. The 2in1 LR reaction were performed with destination vector pBiFCt-2in1-CC and different pDONR221 vectors[33]. Agrobacterial suspensions containing *p35S:PIF4-nYFP-p35S:RFP-p35S:BZR1-cYFP*, *p35S:PIF4-nYFP-p35S:RFP-p35S:BZR1*$^{5S}$*-cYFP*, *p35S:ARF6-nYFP-p35S:RFP-p35S:BZR1-cYFP*, or *p35S:ARF6-nYFP-p35S:RFP-p35S:BZR1*$^{5S}$*-cYFP* constructs were injected into the lower epidermis of tobacco leaves. The transfected plants were kept in the greenhouse for at least 36 h at 22 ℃, and then treated with mock solution or 1 mM $H_2O_2$ for 30 min. Fluorescent signals were visualized by using the LSM-700 laser scanning confocal microscope (Zeiss) and the signal intensities of YFP and RFP were determined by ImageJ software.

**RNA-Seq**. Wild-type Col-0 and *bzr1-1D* were grown on ½ MS medium containing 2 µM PPZ, and/or 1 µM DPI, 0.3 mM $H_2O_2$ for 6 days in the dark. Total RNA was extracted with Trizaol RNA extraction kit (Transgene), and the mRNA sequencing libraries construction and sequencing on the BGISEQ-500 platform were performed at Beijing Geonomics Institute. The sequence reads were mapped to the Arabidopsis genome using HISAT and Bowtie2 software, and differential gene expression was analyzed using Noiseq software. Differentially expressed genes were defined by a 1.5-fold expression difference with a possibility >0.8. The accession number for the RNA-Seq data in the Gene Expression Omnibus database is GSE110488.

**Data availability**. All sequencing data that support the findings of this study have been deposited in the National Center for Biotechnology Information Gene Expression Omnibus (GEO) and are accessible through the GEO Series accession number GSE110488. All data relevant to this study are provided in the manuscript and its supplementary files or are available from the corresponding author upon request.

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

## Acknowledgements

We acknowledge Prof. Chong Kang and Prof. Jianru Zuo for insightful comments on the manuscript; Prof. Yan Guo for providing the seeds of *CAT2-Ox* and *CAT3-Ox* transgenic plants; and Prof. Jianmin Li for providing the construct of *p35S:BZR1ΔDM-YFP*. We thank Jing Zhu and Zhifeng Li from Analysis and Testing Center of SKLMT (State Key Laboratory of Microbial Technology, Shandong University) for assistance in liquid chromatography-mass spectrometry. This work was supported by grants from the National Natural Science Foundation of China (grant no. 31470376, 31670284, and 31600199), the Shandong Province Natural Science Foundation (grant no. JQ201708), and the Ministry of Science and Technology of China (grant no. 2013CB967300).

## Author contributions

Y.T., M.F., and M.-Y.B. together designed the experiments. Y.T. performed statistical analysis of plant growth, oxidation assay, transient expression assay, pull-down assay, and rBiFC. M.F. and Y.T. carried out RNA-Seq experiments and did GO analysis. Z.Q. performed the statistical analysis of QC cell division, and generated *TRXh5-Ox/bzr1-1D*. Z.Q., Y.T., and H.L. analyzed the content of $H_2O_2$ in wild-type plants and BR-related mutants with or without different chemical treatments. X.L., M.W., W.Z., and N.Z. help to generate different BZR1 mutation transgenic plants and rBiFC constructs. Y.T., Z.Z., C.H., and W.W. performed the Mass spec analysis. Z.D. and Z.-Y.W. provided the critical discussion on the work and edited the manuscript. Y.T. performed all other experiments. M.F. and M.-Y.B. wrote the manuscript.

## Additional information

**Competing interests:** The authors declare no competing interests.

