## [Peer Review File · Nature Communications]

Reviewers' comments:

Reviewer #1 (Remarks to the Author):

The manuscript by Bai and colleagues explores the crosstalk involved in brassinosteroid and hydrogen peroxide signalling through the redox regulatory mechanisms of the transcription factor BRASSINAZOLE-RESISTANT1 (BZR1).

The main arguments are the phenotypes and biochemical evidences observed using different inhibitors and loss- and gain-of-function mutants, providing insights into the redox regulation of BZR1.

Although the results are sound the manuscript is still preliminar and has some flaws presented below that must be addressed.

The major concern is the need to definitely demonstrate the physiological meaning of the H₂O₂ induced interaction of BZR1 and its partners PIF4/ARF6. Is the binding to DNA of BZR1-PIF4 and BZR1-ARF6 increased or decreased in the presence of H₂O₂ and TRXh5, respectively?

Are specific target genes altered that could be analyzed by ChIP seq rather than just by RNA seq?

The writing of the paper is for most parts not so clear. There are many grammatical mistakes and most parts of the text are difficult to read and need rephrasing. For instance, 0. and 0, are used in figures. This should be corrected throughout the manuscript.

In addition to these major concerns the paper could be improved for some aspects:

1. Using chemical treatments and genetic tools that decrease the level of H₂O₂, the authors work suggests that H₂O₂ is required for the BR promotion of cell elongation observed during oxidative stress. However, the use of these and other available genetic tools to alter the H₂O₂ levels (i.e. RBOH and catalase mutants, catalase overexpressors) would be more reliable to analyze the in vivo oxidation of BZR1.

MS analysis of BZR1 Cys modifications would definitely confirm the oxidation of the corresponding residues. How can the authors discard by biotin switch assays that the Cys modifications are produced by H₂O₂ and not by any other oxidizing compound?

2. The induced production of H₂O₂ by BL is detected by fluorescent dyes in Supplementary Figure 1 and requires more attention. Here, the fluorescent intensity is very different between mock Col-0 in b and d pannels.

Indeed, authors claim the BR-induced quiescent center (QC) cell divisions are altered by H₂O₂ accumulation. To this end, the detection of H₂O₂ in the QC should be of great relevance to alter this phenotype and this seems not to be the case.

For a better understanding, graphs c and d should represent the same parameters

According to this idea, the localization of H₂O₂ should be close to their corresponding targets (BZR1 in this work). Co-localization experiments would help to understand this fact.

3. In Figure 1 c and d, the relative lenght of the Col-0 hypocotyls is quite different for BL concentrations of 5 nM. These results should be refined.

Also in Figure 1 legend (e,f) bi2bil1bil2 mutant is used while it is not displayed in the corresponding pannels.

4. The relevance of BZR1-PIF4 and BZR1-ARF6 interactions should be considered equally in terms of including the results in the body text and in the supplementary information. Figures should be modified accordingly.

5. The identification of thioredoxin protein TRXh5 as a redox modulator of BZR1 is an interesting finding. However, why this protein was specifically selected? Were other members of the TRX-h family also tested, as commented in page 13 (line 25)?

Interestingly, TRXh5 has been recently reported by Spoel and colleagues to function during plant immunity (Selective protein denitrosylation activity of Thioredoxin-h5 modulates plant Immunity. Mol Cell. 2014 Oct 2;56(1):153-62. doi: 10.1016/j.molcel.2014.08.003).

Thus, how is explained the specificity of the interaction also with a BR signalling master regulator during developmental processes? This reference should be included and the discussion modified accordingly.

In vivo evidences for TRXh5 and BZR1 interaction would help to clarify this point.

Again, the interaction of TRXh5 and BZR1 means both proteins (and the corresponding genes) should be present in the same tissues or developmental stages. This fact should also be considered.

6. A model showing the role for H₂O₂ in the regulation of BZR1 and its partners during BR signalling would help to highlight the findings of this work.

Reviewer #2 (Remarks to the Author):

The work by Tian et al. reported the redox regulation of brassinosteroid (BR) signalling in Arabidopsis, particularly on the BR-activated transcription factor BZR1. The results are quite interesting. They showed by in vitro and in vivo studies that H₂O₂ induces oxidation of BZR1 and the oxidative modification of BZR1 enhances its transcriptional activity by promoting its interaction with key transcription regulators from auxin- and light-signaling pathways, such as ARF6 and PIF4. Genome-wide analysis defined a H₂O₂-dependent BZR1-regulated transcriptome and revealed that the transcriptional activity of BZR1 was dependent on its oxidative condition. Further, they identified the cys-63 residue as the conserved oxidative site in BZR1 (cys-84 in BES1) and mutagenesis of the oxidative site significantly attenuated BZR1's function in regulating gene expression and different biological processes including hypocotyl cell elongation and QC cell division in roots. Furthermore, they identified the thioredoxin protein TRXh5 as the switch that catalyzes BZR1 from oxidation form to reduction form through direct protein-protein interaction. These findings are novel and represent an important advance in our understanding of BR mechanisms in regulating plant growth and development. Based on Tian et al.'s results, the transcription factor BZR1 acts as an oxidative target that bridges the H₂O₂-mediated redox signalling and BR signaling pathways in seedling development.

Although the manuscript is well written and the experiments are well designed and done, I have the following points that require the authors to consider when revise and improve their manuscript.

Major ones:

1. In Fig.3A & B, the authors said that the transgenic plants of *bzr1-ID*^{C91,174,240S}-Ox have similar growth and PPZ-insensitive phenotypes to *bzr1-ID*-Ox, whereas *bzr1-ID*^{C63S}-Ox, *bzr1-ID*^{DC63,73S}-Ox and *bzr1-ID*^{5S}-Ox transgenic plants display no curled-leaves and normal sensitivity to PPZ, similar to wild type plants. This description might be not accurate. It appears from Fig. 3A & B that although *bzr1-ID*^{C63S}-Ox, *bzr1-ID*^{DC63,73S}-Ox and *bzr1-ID*^{5S}-Ox plants have reduced insensitivity to PPZ compared to *bzr1-ID*-Ox, their insensitivity to PPZ is still higher than the wild type plants.
2. In the transient assay of BZR1 transcriptional activity in Fig. 3h, the authors used a LUC reporter system driven by a G-box promoter and found that *bzr1-ID* repressed G-box-driving LUC expression. I do not understand why BZR1 inhibited G-box promoter-driving gene expression and why the authors did not use a specific target gene(s) of BZR1 to do the experiments so that the result will be more meaningful. The same experiment was done in Fig. 6h but here BZR1 promoted the LUC expression. It is not known whether the same G-box promoter was used here and why it gives opposite result as in Fig. 3h?
3. In the transcriptome profiling of H₂O₂-regulated genes, why the authors used the ROS scavenger PPI to do the treatment instead of using H₂O₂ directly, which may be more straight forward to understand? Is the chemical PPI specific to H₂O₂ and has no any other effects on plants such as growth?
4. Page11: The authors stated that the genes involved in cell growth and auxin responses were highly enriched in H₂O₂-dependent BZR1-activated genes, but the genes involved in ROS production, photosynthesis and chloroplast were highly enriched in H₂O₂-dependent BZR1-repressed genes. It will be nice to discuss/explain why genes

involved in ROS production, photosynthesis and chloroplast were more enriched in H₂O₂-dependent BZR1-repressed genes?

5. Fig. 6: The authors showed the QC cell division rate between Col-0 and the *TRXh5-Ox* and its weak suppression to *bzr1-1D*'s insensitivity to the BR biosynthesis inhibitor PPZ. It is better to also show the growth phenotypes of *TRXh5-Ox* plants in both light and dark conditions, so people will better understand the function of TRXh5 in redox regulation of BZR1 and plant development.

Minor ones:

1. Some of the figure legends are too simply written. For example, in the legends of Fig. 2, the authors only described the results of each experiment but did not explain how the experiments were done. Key information of experiments must be given in order the readers to better understand the figure. This is also true for Fig. 6h. In addition, in Fig. 2C, there is a label of MPH-BES1^{C84S} but it is not known what MPH is. It should be explained in figure legend.
2. Page 9, lines 8-11 and Page 10, lines 10-11: All the *bzr1-1D* should be italic.

Reviewer #3 (Remarks to the Author):

This is a very exciting and novel manuscript that describes the mechanism by which H₂O₂ and BR interact to regulate proliferation and elongation of plant cells. The authors identified the transcriptional regulator BZR1 as a target for H₂O₂ modification on Cys 63 and conducted extensive analysis of how this modification alters protein-protein interactions and transcription of thousands of transcripts in the cell. Moreover, the authors identified an h-type TRX as involved in regulating the H₂O₂-induced oxidation of BZR1 in vitro and in vivo.

In my opinion this is a highly important and novel work that is relevant to a wide audience and merits publication in a high impact journal.

Although the work reported in this manuscript is highly important and novel, its presentation requires major additional work as highlighted below:

1. This manuscript requires extensive editing with special attention to the use of English. There are way too many problems in way too many places for me to even start correcting or highlighting them.
2. Please mention the SA-NPR-redox work (and any other H₂O₂-hormone related work) in the introduction. As presented in the introduction it seems that this work was the first to report ROS-regulation of hormone responses.
3. There appears to be a problem with many of the references (e.g., 17, 18). Please correct
4. The experiment in Figs 1e and 1f needs to be better explained. I am not sure why an inhibitor of BR signaling was used? Also, does the *bin2* triple mutants has high levels of ROS in the presence or absence of this inhibitor? Regarding this point, if the statement "the involvement of H₂O₂ in BR signaling pathway should be independent on BIN2 and its homolog proteins" is correct, why show the data for the *bin2* triple mutant?
5. I think it is an example of the problem with English, but how was "Reagent H₂O₂ was removed by

precipitation" in the protein oxidation assay? Also, regarding this assay, the authors used 1mM H₂O₂ for the in vitro assay. This seems to me to be a very high concentration of H₂O₂ that is possibly not physiological. Did the authors use lower concentrations? Did the work?

6. Please indicate in Fig. 2d what is the double band shown in "Total BZR1" under control "M" conditions.

7. Figure 3a needs better labeling. Please indicate all treatments/lines.

8. Figure 4b does not look right. Are there no genes in the DPI lane? Also, regarding the RNA-seq experiment, for all the treatments used for this assay plants were incubated with the different inhibitors/BR for 6 days in the dark??? Can the authors explain why this method was chosen?

9. Quantification and statistical analysis are required for Figs. 5a and 5g. This is especially critical for Fig. 5g, since this is the only real evidence for in vivo interaction that is enhanced by H₂O₂. In fact this is the main weakness of the paper. I understand that Fig. 5i is statistical analysis, but it is only for 20 cells from transient expression in tobacco by infiltration with agro. We need to see that data from Arabidopsis and preferably from transgenic plants.

10. What is "BES1-1 cYFP" (Page 14 lane 1)?

11. Quantification and statistical analysis are required for Figs. 6a

12. A model figure is highly needed for this paper.

Responses to comments by reviewers

We wish to express our deep appreciation for the constructive comments on our manuscript by the reviewers. In response to these comments, we have conducted additional experiments and modified the text extensively to improve our manuscript. Specifically, we have added the following results:

1. We have performed the BIAM labeling assay (Fig. 2c) and LC-MS/MS experiment (Fig. 2e) to confirm the oxidative modification of BZR1 by H₂O₂.
2. We have performed co-immunoprecipitation (CoIP) assays to confirm that H₂O₂ promotes the interactions of BZR1 with PIF4 (Fig. 5j) and ARF6 (Supplementary Fig. 8g) in plants.
3. We have performed CoIP assay to confirm the interaction between BZR1 and TRXh5 in plants (Fig. 6c).
4. We have added a model for the regulatory network of BR and H₂O₂ in cell elongation (Fig. 7).
5. We have performed the BES1 experiments to show that the H₂O₂-oxidized cys-84 in BES1 is required for its function in cell elongation (Supplementary Fig. 5a-f).
6. We have added the DNA-protein pull-down and chromatin immunoprecipitation quantitative PCR (ChIP-qPCR) experiments to show that H₂O₂ has no significant effects on the DNA-binding ability of BZR1 *in vivo* and *in vitro* (Supplementary Fig. 7a-d).
7. We have added data showing *TRXh5* overexpression attenuated the growth phenotype of *bzr1-1D* (Supplementary Fig. 11a-g).
8. We have added RT-qPCR and ChIP-qPCR to show that BR represses the expression of *TRXh5* through BZR1 (Supplementary Fig. 12a,b).
9. We have added data showing the similar distribution patterns of H₂O₂ and nuclear BZR1 in root tips (Supplementary Fig. 13a,b).

Please find below a detailed response to the points raised.

Sincerely,

Mingyi

Reviewers' comments:

Reviewer #1 (Remarks to the Author):

The manuscript by Bai and colleagues explores the crosstalk involved in brassinosteroid and hydrogen peroxide signalling through the redox regulatory mechanisms of the transcription factor BRASSINAZOLE-RESISTANT1 (BZR1). The main arguments are the phenotypes and biochemical evidences observed using different inhibitors and loss- and gain-of-function mutants, providing insights into the redox regulation of BZR1. Although the results are sound the manuscript is still preliminar and has some flaws presented below that must be addressed. The major concern is the need to definitely demonstrate the physiological meaning of the H₂O₂ induced interactions of BZR1 and its partners PIF4/ARF6. Is the binding to DNA of BZR1-PIF4 and BZR1-ARF6 increased or decreased in the presence of H₂O₂ and TRXh5, respectively? Are specific target genes altered that could be analyzed by ChIP seq rather than just by RNA seq?

Response: Thanks for this excellent suggestion. To determine the physiologic meaning of the H₂O₂ induced interaction of BZR1 with PIF4 and ARF6, we have performed DNA-protein pull-down assay, ChIP-qPCR assay and protoplast transient gene expression assay to analyze whether H₂O₂ alters the DNA binding ability and/or transcriptional activity of BZR1. Our results showed that H₂O₂ treatment did not enhance the *in vitro* binding activity of BZR1 to the promoter of *SAUR15* in the absence or presence of PIF4 and ARF6 (Supplementary Fig. 7a-c). ChIP-qPCR analysis showed that H₂O₂ did not have any significant effect on BZR1 DNA-binding capacity in plants (Supplementary Fig. 7d), suggesting that H₂O₂ does not affect the DNA binding ability of BZR1. In contrast, H₂O₂ significantly improved the transcriptional activity of BZR1, PIF4 and ARF6 in protoplast transient gene expression assays (Supplementary Fig. 7e). In addition, GO comparison of BZR1-regulated genes under oxidized and reduced states showed that the BZR1-activated H₂O₂-dependent genes are enriched in PIF4- or ARF-bound/regulating genes, which is consistent with the recent finding that auxin, light, and BR regulate shoot cell elongation through a central module of BZR1 interactions with the PIF4 and ARF6 factors (Supplementary Fig. 6a,b). These results support that H₂O₂ promotes the interactions of BZR1 with PIF4 and ARF6, instead of affecting BZR1-DNA binding, to enhance the target gene expression.

The writing of the paper is for most parts not so clear. There are many grammatical mistakes and most parts of the text are difficult to read and need rephrasing. For instance, 0. and 0, are used in figures. This should be corrected throughout the manuscript.

Response: Thank you for pointing this out. We have extensively edited the revised manuscript.

In addition to these major concerns the paper could be improved for some aspects:

1. Using chemical treatments and genetic tools that decrease the level of H₂O₂, the authors work suggests that H₂O₂ is required for the BR promotion of cell elongation observed during oxidative stress. However, the use of these and other available genetic tools to alter the H₂O₂ levels (i.e. RBOH and catalase mutants, catalase overexpressors) would be more reliable to analyze the in vivo oxidation of BZR1.

Response: As suggested, we have analyzed the oxidation of BZR1 in CAT2 overexpression transgenic plants. The results show that the level of oxidized BZR1 is significantly decreased in the CAT2-Ox plants, suggesting that the oxidation level of BZR1 is closely related to the endogenous level of H₂O₂ in plants (Fig. 2i).

MS analysis of BZR1 Cys modifications would definitely confirm the oxidation of the corresponding residues.

Response: Thanks for the suggestion. Indeed, we analyzed tryptic fragments derived from the recombinant BZR1 protein sequentially treated with H₂O₂, NEM, and BIAM by liquid chromatography-tandem mass spectrometry, and identified cys-63, cys-91, cys-174 and cys-240 as oxidized residues (Fig. 2e). Since mutation of cys-63 significantly reduced the oxidative modification of BZR1 by H₂O₂ (Fig. 2f,g), and cys-63 is highly conserved among various species (Supplementary Fig. 4b), we focused on the analysis of cys-63 in this study.

How can the authors discard by biotin switch assays that the Cys modifications are produced by H₂O₂ and not by any other oxidizing compound?

Response: It has been reported that biotin-conjugated iodoacetamide (BIAM) and H₂O₂ selectively and competitively react with cysteine residues that exhibit a low pK_a in target proteins¹. Based on this observation, the techniques including BIAM labeling assay and biotin-switch assay were developed to detect the H₂O₂-sensitive cysteine residues of target protein (Fig. 2a,b). In our previous manuscript, we only used the biotin-switch assay to analyze the oxidative modification of BZR1 by H₂O₂. In an attempt to confirm the reliability of this result, we performed the BIAM labeling assay to analyze the oxidation of BZR1 by H₂O₂. Our results showed that the recombinant protein MBP-BZR1 could be labeled by BIAM and that labeling levels were decreased by H₂O₂ treatment in a dose-dependent manner, suggesting H₂O₂ causes oxidation of BZR1 *in vitro* (Fig. 2c).

2. The induced production of H₂O₂ by BL is detected by fluorescent dyes in Supplementary Figure 1 and requires more attention. Here, the fluorescent intensity is very different between mock Col-0 in b and d panels.

Response: We have repeated these experiments and new photographs were taken with the same parameters (Supplementary Fig. 1b and e).

Indeed, authors claim the BR-induced quiescent center (QC) cell divisions are altered by H₂O₂ accumulation. To this end, the detection of H₂O₂ in the QC should be of great relevance to alter this phenotype and this seems not to be the case.

Response: We have carefully analyzed the levels of H₂O₂ in the QC and found that BR significantly promoted the accumulation of H₂O₂ in the QC area, whereas BR and DPI co-treatment reduced H₂O₂ content (Supplementary Fig. 1b and d). BR-triggered enrichment of H₂O₂ in the QC would induce the oxidation of BZR1 to increase its transcriptional activity and then reprogram target gene expression, and promote QC cell division.

For a better understanding, graphs c and d should represent the same parameters.

Response: We have repeated these experiments and new photographs were taken with the same parameters

According to this idea, the localization of H₂O₂ should be close to their corresponding targets (BZR1 in this work). Co-localization experiments would help to understand this fact.

Response: As suggested by the reviewer, we have analyzed the distribution of BZR1 and H₂O₂ in the root tips (Supplementary Fig. 13a,b). The results showed that both BZR1 and H₂O₂ exhibit gradient distribution patterns in the Arabidopsis root, consistent with previous reports^{2, 3}. Low levels of nuclear-localized BZR1 are observed in the stem cell niche, whereas high levels of BZR1 are observed in the nucleus of the epidermal cells in the transition-elongation zone to promote cell elongation. H₂O₂ also accumulates mainly in the expanding cells of the elongation zone to promote cellular differentiation, whereas, superoxide mainly accumulates in the dividing cells of the meristem zone to maintain stem cell. Overall, the distribution pattern of H₂O₂ is similar to that of nuclear BZR1 (Supplementary Fig. 13a,b), consistent with H₂O₂ modulating BZR1 activity in root growth.

3. In Figure 1 c and d, the relative length of the Col-0 hypocotyls is quite different for BL concentrations of 5 nM. These results should be refined.

Response: We have repeated these experiments under the same conditions and made new graphs (Fig. 1c,d).

Also in Figure 1 legend (e,f) *bi2bil1bil2* mutant is used while it is not displayed in the corresponding panels.

Response: We have corrected the legend for Figure 1.

4. The relevance of BZR1-PIF4 and BZR1-ARF6 interactions should be considered equally in terms of including the results in the body text and in the supplementary

information. Figures should be modified accordingly.

Response: We agree and therefore we have changed the manuscript as suggested.

5. The identification of thioredoxin protein TRXh5 as a redox modulator of BZR1 is an interesting finding. However, why this protein was specifically selected? Were other members of the TRX-h family also tested, as commented in page 13 (line 25)?

Response: Thank you for pointing this out. We had performed the BiFC assays to detect the interactions between BZR1 and members of TRX-h family in tobacco leaves. Results showed that TRXh5 specifically interacted with BZR1 in the epidermal cells of tobacco leaves (Supplementary Fig. 9).

Interestingly, TRXh5 has been recently reported by Spoel and colleagues to function during plant immunity (Selective protein denitrosylation activity of Thioredoxin-h5 modulates plant Immunity. Mol Cell. 2014 Oct 2;56(1):153-62. doi: 10.1016/j.molcel.2014.08.003). Thus, how is explained the specificity of the interaction also with a BR signalling master regulator during developmental processes? This reference should be included and the discussion modified accordingly.

Response: We have included this reference in our text and added the discussion as followed:

“Detailed characterization of TRXh5 in this study reveals that the growth-promoting hormone BR represses the expression of *TRXh5* through BZR1 (Supplementary Fig. 12a,b) and that TRXh5 negatively regulates BR signal transduction by catalyzing BZR1 oxidation-to-reduction switch. Previous studies have shown that TRXh5 regulates SA-dependent plant immunity by selective protein denitrosylation^{4, 5}. Pathogens and SA highly upregulate the expression of *TRXh5*, which reduces NPR1 to cause its oligomer-monomer exchange and nuclear localization to activate immune responses. As such, TRXh5 is activated by pathogens but repressed by BR, and it activates immunity and represses growth, and thereby functions as a major node of crosstalk that mediates a trade-off between growth and immunity in plants.”

In vivo evidences for TRXh5 and BZR1 interaction would help to clarify this point. Again, the interaction of TRXh5 and BZR1 means both proteins (and the corresponding genes) should be present in the same tissues or developmental stages. This fact should also be considered.

Response: We have analyzed the expression patterns of *BZR1* and *TRXh5* using microarray data (eFP browser, <http://bar.utoronto.ca/efp/cgi-bin/efpWeb.cgi>)⁶ and found very similar ubiquitous expression patterns of *TRXh5* and *BZR1* in plants (Supplementary Fig. 10a,b). To determine the interaction between TRXh5 and BZR1 in plants, we have

performed co-immunoprecipitation (CoIP) assays using Arabidopsis transgenic plants expressing *BZR1-YFP* and *TRXh5-Myc* (Fig 6c); results showed TRXh5 interacts with BZR1 in plants.

6. A model showing the role for H₂O₂ in the regulation of BZR1 and its partners during BR signalling would help to highlight the findings of this work.

Response: We have added a model in Figure 7 to explain how H₂O₂ and thioredoxin antagonistically regulate the activity of BZR1 to modulate the BR signal transduction..

Reviewer #2 (Remarks to the Author):

The work by Tian et al. reported the redox regulation of brassinosteroid (BR) signalling in Arabidopsis, particularly on the BR-activated transcription factor BZR1. The results are quite interesting. They showed by in vitro and in vivo studies that H₂O₂ induces oxidation of BZR1 and the oxidative modification of BZR1 enhances its transcriptional activity by promoting its interaction with key transcription regulators from auxin- and light-signaling pathways, such as ARF6 and PIF4. Genome-wide analysis defined a H₂O₂-dependent BZR1-regulated transcriptome and revealed that the transcriptional activity of BZR1 was dependent on its oxidative condition. Further, they identified the cys-63 residue as the conserved oxidative site in BZR1 (cys-84 in BES1) and mutagenesis of the oxidative site significantly attenuated BZR1's function in regulating gene expression and different biological processes including hypocotyl cell elongation and QC cell division in roots. Furthermore, they identified the thioredoxin protein TRXh5 as the switch that catalyzes BZR1 from oxidation form to reduction form through direct protein-protein interaction. These findings are novel and represent an important advance in our understanding of BR mechanisms in regulating plant growth and development. Based on Tian et al.'s results, the transcription factor BZR1 acts as an oxidative target that bridges the H₂O₂-mediated redox signalling and BR signaling pathways in seedling development. Although the manuscript is well written and the experiments are well designed and done, I have the following points that require the authors to consider when revise and improve their manuscript.

Major ones:

1. In Fig.3A & B, the authors said that the transgenic plants of *bzr1-1D*^{C91,174,240S}-Ox have similar growth and PPZ-insensitive phenotypes to *bzr1-1D*-Ox, whereas *bzr1-1D*^{C63S}-Ox, *bzr1-1D*^{C63,73S}-Ox and *bzr1-1D*^{5S}-Ox transgenic plants display no curled leaves and normal sensitivity to PPZ, similar to wild type plants. This description might be not accurate. It appears from Fig. 3A & B that although *bzr1-1D*^{C63S}-Ox, *bzr1-1D*^{C63,73S}-Ox and *bzr1-1D*^{5S}-Ox plants have reduced insensitivity to PPZ compared to *bzr1-1D*-Ox, their

insensitivity to PPZ is still higher than the wild type plants.

Response: We have changed the sentence as follows:

“whereas *bzr1-1D*^{C63S}, *bzr1-1D*^{C63,73S} and *bzr1-1D*^{5S} transgenic plants all displayed normal leaf phenotypes and increased sensitivity to PPZ (Fig. 3a-e)”

2. In the transient assay of BZR1 transcriptional activity in Fig. 3h, the authors used a LUC reporter system driven by a G-box promoter and found that *bzr1-1D* repressed G-box-driving LUC expression. I do not understand why BZR1 inhibited G-box promoter-driving gene expression and why the authors did not use a specific target gene(s) of BZR1 to do the experiments so that the result will be more meaningful. The same experiment was done in Fig. 6h but here BZR1 promoted the LUC expression. It is not known whether the same G-box promoter was used here and why it gives opposite result as in Fig. 3h?

Response: Thank you for pointing this out. We have used the *PRE5* promoter in all transient gene expression assays in this MS, and found *bzr1-1D* significantly increased the expression of luciferase reporter driven by the *PRE5* promoter, whereas *bzr1-1D*^{C63S} only marginally increased *PRE5* promoter activity, and *bzr1-1D*^{5S} lost the transcriptional activation activity (Fig. 3h).

3. In the transcriptome profiling of H₂O₂-regulated genes, why the authors used the ROS scavenger DPI to do the treatment instead of using H₂O₂ directly, which may be more straight forward to understand?

Response: BR treatment not only induces dephosphorylation of BZR1, but also triggers the production of H₂O₂, which in turn will promote oxidation of BZR1. Phosphorylation-dephosphorylation is the key switch to control the activity of BZR1, while oxidation-reduction works mainly to fine-tune the activity of BZR1. These two regulatory mechanisms confer an efficient control of BZR1 activity to regulate downstream genes expression. Under normal conditions, endogenous BR level is low and most of BZR1 have no transcriptional activity, due to phosphorylation by GSK3 kinase BIN2. At this point, addition of H₂O₂ has no clear effect on the activity of BZR1. However, in the case of BR treatment, we added DPI to reduce the content of H₂O₂ in plants, thereby making it easier to detect the effect of oxidation on the transcription activity of BZR1.

Is the chemical DPI specific to H₂O₂ and has no any other effects on plants such as growth?

Response: DPI is a specific NADPH oxidase inhibitor that markedly diminished the production of H₂O₂ derived from diverse plant physiologic processes, such as stem cell differentiation⁷, root growth³ and stomatal movement⁸. Here, we showed that DPI reduced

the transcriptional activity of BZR1 to regulate cell elongation and QC cell division. To examine the true effect of DPI on the BZR1 transcriptional activity, we added back H₂O₂ to DPI-treated plants. Results showed that under an H₂O₂-deficient condition, the amplitude of BZR1-regulated gene expression was significantly reduced and approximately 64% of the genes were no longer altered in *bzr1-1D*; while adding H₂O₂ back to the H₂O₂-deficient condition partially restored the responsiveness to *bzr1-1D* for about 58% of the BZR1-regulated genes. On the other hand, 2200 (54%) of the 4054 DPI-altered genes were also altered by *bzr1-1D*, with 93.3% of them affected oppositely by DPI and *bzr1-1D*. Such strong correlation between DPI and BZR1 effects of transcriptome supports that BZR1 is a major factor for H₂O₂ modulation of gene expression under our experimental conditions.

4. Page11: The authors stated that the genes involved in cell growth and auxin responses were highly enriched in H₂O₂-dependent BZR1-activated genes, but the genes involved in ROS production, photosynthesis and chloroplast were highly enriched in H₂O₂-dependent BZR1-repressed genes. It will be nice to discuss/explain why genes involved in ROS production, photosynthesis and chloroplast were more enriched in H₂O₂-dependent BZR1-repressed genes?

Response: As suggested, we have added the discussion as follows:

“H₂O₂-dependent BZR1-repression genes are enriched in photosynthesis and ROS production. Such BZR1 repression of ROS-related genes may contribute to feedback regulation and redox homeostasis.”

5. Fig. 6: The authors showed the QC cell division rate between Col-0 and the *TRXh5-Ox* and its weak suppression to *bzr1-1D*'s insensitivity to the BR biosynthesis inhibitor PPZ. It is better to also show the growth phenotypes of *TRXh5-Ox* plants in both light and dark conditions, so people will better understand the function of TRXh5 in redox regulation of BZR1 and plant development.

Response: Thank you for pointing this out. We have analyzed the growth phenotype of wild type Col-0, *bzr1-1D*, *TRXh5-Ox* and *TRXh5-Ox/bzr1-1D* in both light and dark conditions. Results showed that *TRXh5* overexpression not only partly restored the sensitivity of *bzr1-1D* to PPZ in the dark, but also attenuated the large leaves of *bzr1-1D* under light (Fig. 6h and Supplementary Fig. 11a-g), indicating TRXh5 negatively regulated the activity of BZR1.

Minor ones:

1. Some of the figure legends are too simply written. For example, in the legends of Fig. 2, the authors only described the results of each experiment but did not explain how the

experiments were done. Key information of experiments must be given in order the readers to better understand the figure. This is also true for Fig. 6h. In addition, in Fig. 2C, there is a label of MPH-BES1^{C84S} but it is not known what MPH is. It should be explained in figure legend.

Response: Thank you for pointing this out. We have provided the missing details in the revised manuscript.

2. Page 9, lines 8-11 and Page 10, lines 10-11: All the *bzr1-1D* should be italic.

Response: Thank you, we have made the change, as suggested.

Reviewer #3 (Remarks to the Author):

This is a very exciting and novel manuscript that describes the mechanism by which H₂O₂ and BR interact to regulate proliferation and elongation of plant cells. The authors identified the transcriptional regulator BZR1 as a target for H₂O₂ modification on Cys 63 and conducted extensive analysis of how this modification alters protein-protein interactions and transcription of thousands of transcripts in the cell. Moreover, the authors identified an h-type TRX as involved in regulating the H₂O₂-induced oxidation of BZR1 *in vitro* and *in vivo*. In my opinion, this is a highly important and novel work that is relevant to a wide audience and merits publication in a high impact journal. Although the work reported in this manuscript is highly important and novel, its presentation requires major additional work as highlighted below:

1. This manuscript requires extensive editing with special attention to the use of English. There are way too many problems in way too many places for me to even start correcting or highlighting them.

Response: Thank you for pointing this out, we have extensively edited the revised manuscript.

2. Please mention the SA-NPR-redox work (and any other H₂O₂-hormone related work) in the introduction. As presented in the introduction it seems that this work was the first to report ROS-regulation of hormone responses.

Response: We have added a paragraph in the introduction to describe the relationship between H₂O₂ and various hormones.

3. There appears to be a problem with many of the references (e.g., 17, 18). Please correct.

Response: Thank you. We have fixed them.

4. The experiment in Figs 1e and 1f needs to be better explained. I am not sure why an inhibitor of BR signaling was used?

Response: To determine the correct effect of *bzr1-1D* on cell elongation, the BR biosynthesis inhibitor PPZ was generally used to reduce BR endogenous content, which caused inactivation of wild-type BZR1 and its homolog, but not *bzr1-D* proteins. Here, we showed that the insensitivity of *bzr1-1D* to PPZ was eliminated by DPI treatment, suggesting DPI-mediated production of H₂O₂ is required for BZR1 function.

Also, does the *bin2* triple mutants has high levels of ROS in the presence or absence of this inhibitor? Regarding this point, if the statement “the involvement of H₂O₂ in BR signaling pathway should be independent on BIN2 and its homolog proteins” is correct, why show the data for the *bin2* triple mutant?

Response: Hydrogen peroxide (H₂O₂) and Nitric oxide (NO) are two key signaling molecules in the redox signaling realm and both can react with certain reactive cysteine residues to induce oxidative modification of target proteins⁹. H₂O₂ and NO sometimes share common target proteins to transduce redox signals. For example, NPR1, the master regulator in plant immunity, is induced to occur oxidative modification by both H₂O₂ and NO and then undergoes polymerization and loss of its activity^{5,10}. It has been reported that NO-induced S-nitrosylation inhibited the kinase activity of BIN2 *in vitro*¹¹. To determine whether H₂O₂ regulates the BR signal transduction through BIN2, we examined the H₂O₂ effect on cell elongation and QC cell division in *bin2* triple mutants and wild type plants treated with GSK3 inhibitors bikinin or LiCl. Results showed that, although BIN2 and its homolog proteins completely lose their activity, H₂O₂ was able to regulate cell elongation and QC cell division, indicating that H₂O₂ contributes to BR signal transduction by targeting the downstream components of BIN2.

5. I think it is an example of the problem with English, but how was “Reagent H₂O₂ was removed by precipitation” in the protein oxidation assay?

Response: Thank you for pointing this out, we have changed the sentence as followed:

“The proteins were precipitated by adding one volume of acetone at -20°C for 20 min and centrifuged at 5000 g for 5 min. The pellet was washed 3 times with 50% acetone and dissolved in 500 µl EBR buffer.”

Also, regarding this assay, the authors used 1mM H₂O₂ for the in vitro assay. This seems to me to be a very high concentration of H₂O₂ that is possibly not physiological. Did the authors use lower concentrations? Did the work?

Response: As suggested, we have detected the oxidation of BZR1 by incubation with different concentrations of H₂O₂. Results showed that H₂O₂ promoted oxidative

modification of BZR1 in a dose-dependent manner (Fig. 2d)

6. Please indicate in Fig. 2d what is the double band shown in “Total BZR1” under control “M” conditions.

Response: We have added the labeling for the double bands shown in Figure 2h (that is the original Fig. 2d)

7. Figure 3a needs better labeling. Please indicate all treatments/lines.

Response: We have added the labeling for all treatments and lines in Figure 3a.

8. Figure 4b does not look right. Are there no genes in the DPI lane?

Response: By comparing the transcription profiles of *bzr1-1D* and wild type Col-0, we identified 4428 different genes expressed (DGE) under normal condition, 2458 DGE under the DPI-treated H₂O₂ deficient condition, and 5152 DGE under DPI and H₂O₂ co-treatment condition. To determine the effect of H₂O₂ on the transcriptional activity of BZR1, we analyzed the expression data of 4428 genes regulated by BZR1 under three different conditions. Results showed that BZR1 significantly reduced the magnitude of regulation of gene expression under the DPI condition, but partly recovered when H₂O₂ was added back, indicating H₂O₂ plays crucial roles for the transcriptional activity of BZR1.

Also, regarding the RNA-seq experiment, for all the treatments used for this assay plants were incubated with the different inhibitors/BR for 6 days in the dark??? Can the authors explain why this method was chosen?

Response: As a specific BR biosynthesis inhibitor, propiconazole (PPZ) causes de-etiolated and dwarf phenotypes similar to those of BR-deficient mutants. The dominant mutant *bzr1-1D* is insensitive to PPZ and fails to de-etiolate in a medium containing PPZ. However, in our study we showed that the insensitivity of *bzr1-1D* to PPZ was eliminated by DPI treatment, but was partially restored after H₂O₂ addition back to the medium. By comparing the transcription profile of *bzr1-1D* and wild type plants under conditions including only PPZ treatment, PPZ and DPI treatment, and PPZ, DPI and H₂O₂ co-treatment, we can learn how H₂O₂ regulates the transcription activity of BZR1 at whole genomic level.

9. Quantification and statistical analysis are required for Figs. 5a and 5g. This is especially critical for Fig. 5g, since this is the only real evidence for in vivo interaction that is enhanced by H₂O₂. In fact this is the main weakness of the paper. I understand that Fig. 5i is statistical analysis, but it is only for 20 cells from transient expression in tobacco by infiltration with agro. We need to see that data from Arabidopsis and preferably from

transgenic plants.

Response: For Fig 5a, we have performed statistical analysis of the ratio of BZR1 in nuclei and cytoplasm under different conditions. Consistent with previous results, DPI treatment had no significant effect on the subcellular localization of BZR1 (Fig. 5b).

To examine the effect of H₂O₂ on the interactions of BZR1 with PIF4 and ARF6 in plants, we firstly performed addition rBiFC assays of BZR1-PIF4 and BZR1-ARF6 interactions in the tobacco leaf epidermal cells and statistically analyzed the fluorescent signal intensity from 50 cells (Fig. 5h,i and Supplementary Fig. 8e,f). Further, we performed co-IP assays to determine the effects of H₂O₂ on the interactions of BZR1 with its partners PIF4 and ARF6 in Arabidopsis protoplast. Results showed consistently that H₂O₂ enhanced the interactions between BZR1 and PIF4 or ARF6 (Fig. 5j and Supplementary Fig. 8g). These results together provide strong evidences that H₂O₂ promotes the interactions of BZR1 with PIF4 and ARF6.

10. What is “BES1-1 cYFP” (Page 14 lane 1)?

Response: Thank you. We have added the description of BES1-cYFP in the text.

11. Quantification and statistical analysis are required for Figs. 6a

Response: Fig 6a shows strong BiFC signals indicating interactions of TRX5 with both BZR1 and BES1, but no signal was detectable, or quantifiable, for the negative control. These experiments have been repeated with similar results. Further more, the interaction between BZR1 and TRXh5 has been confirmed by co-immunoprecipitation (CoIP) assays using transgenic Arabidopsis expressing *BZR1-YFP* and *TRXh5-myc* (Fig. 6c).

12. A model figure is highly needed for this paper.

Response: We have added a model in Figure 7 to explain how H₂O₂ and thioredoxin antagonistically regulate the activity of BZR1 to modulate the BR signal transduction.

References:

1. Kim JR, Yoon HW, Kwon KS, Lee SR, Rhee SG. Identification of proteins containing cysteine residues that are sensitive to oxidation by hydrogen peroxide at neutral pH. *Analytical biochemistry* 2000, 283(2): 214-221.
2. Chaiwanon J, Wang ZY. Spatiotemporal brassinosteroid signaling and antagonism with auxin pattern stem cell dynamics in Arabidopsis roots. *Current biology* 2015, 25(8): 1031-1042.
3. Tsukagoshi H, Busch W, Benfey PN. Transcriptional regulation of ROS controls transition from proliferation to differentiation in the root. *Cell* 2010, 143(4): 606-616.
4. Kneeshaw S, Gelineau S, Tada Y, Loake GJ, Spoel SH. Selective protein denitrosylation activity of Thioredoxin-h5 modulates plant Immunity. *Molecular cell* 2014, 56(1): 153-162.
5. Tada Y, Spoel SH, Pajerowska-Mukhtar K, Mou Z, Song J, Wang C, *et al.* Plant immunity requires conformational changes of NPR1 via S-nitrosylation and thioredoxins. *Science* 2008, 321(5891): 952-956.
6. Winter D, Vinegar B, Nahal H, Ammar R, Wilson GV, Provart NJ. An "Electronic Fluorescent Pictograph" browser for exploring and analyzing large-scale biological data sets. *PloS one* 2007, 2(8): e718.
7. Zeng J, Dong Z, Wu H, Tian Z, Zhao Z. Redox regulation of plant stem cell fate. *The EMBO journal* 2017, 36(19): 2844-2855.
8. Pei ZM, Murata Y, Benning G, Thomine S, Klusener B, Allen GJ, *et al.* Calcium channels activated by hydrogen peroxide mediate abscisic acid signalling in guard cells. *Nature* 2000, 406(6797): 731-734.
9. Baxter A, Mittler R, Suzuki N. ROS as key players in plant stress signalling. *Journal of Experimental Botany* 2014, 65: 1229-1240.
10. Mou Z, Fan W, Dong X. Inducers of plant systemic acquired resistance regulate NPR1 function through redox changes. *Cell* 2003, 113(7): 935-944.
11. Wang P, Du Y, Hou YJ, Zhao Y, Hsu CC, Yuan F, *et al.* Nitric oxide negatively regulates abscisic acid signaling in guard cells by S-nitrosylation of OST1. *Proceedings of the National Academy of Sciences of the United States of America* 2015, 112(2): 613-618.

REVIEWERS' COMMENTS:

Reviewer #1 (Remarks to the Author):

This revised version of the manuscript "Hydrogen peroxide positively regulates brassinosteroid signaling through oxidation of transcription factor BRASSINAZOLE-RESISTANT1" by Bai and colleagues has been significantly edited and modified to address my previous queries and questions, as well as those of the two other reviewers.

I can only state as reviewer 1 that most of my comments were considered and addressed in a smart way, and especially the proposed additional experiments were conducted. Together with the responses to reviewers this improved the manuscript significantly. I have only a few more remarks with regard to the new data:

The procedures for the LC-MS/MS analysis and co-immunoprecipitations are not included in the M&M section.

In Supplementary Figure 1 (b) H2DCFDA staining for H2O2 in the primary root tips of Col-0 treated with BL and/or DPI. Please indicated BL+DPI picture.

In Supplementary Figure 8 a different font is used.

New references should be further considered to improve the discussion of the manuscript, especially the very related paper by Lv B, Tian H, Zhang F, Liu J, Lu S, Bai M, et al. (2018) Brassinosteroids regulate root growth by controlling reactive oxygen species homeostasis and dual effect on ethylene synthesis in Arabidopsis. PLoS Genet 14(1): e1007144.

This is all what I can suggest for further improvement as the authors in my opinion mostly addressed all the issues raised by the reviewers.

Reviewer #2 (Remarks to the Author):

1. The authors have well addressed most of my questions raised in the last review except Q4 (major), which was not sufficiently addressed. The question concerns (in transcriptome analysis) why genes involved in ROS production, photosynthesis and chloroplast were more enriched in H2O2-dependent BZR1-repressed genes. The authors responded that: "H2O2-dependent BZR1-repression genes are enriched in photosynthesis and ROS production. Such BZR1 repression of ROS-related genes may contribute to feedback regulation and redox homeostasis." However, it is not clear from this sentence what feedback regulation will the BZR1-repressed ROS genes contribute to. ROS production? If yes, can the authors give a bit more elaboration on how BZR1 repression of photosynthesis and chloroplast related genes will contribute to this feedback regulation?

2. Related to my minor questions #2, there are still many places in the Ms the *bzr1-1D* and *bes1-D* are not italic.

3. The title: A "the" should be added before "transcription factor BRASSINAZOLE-RESISTANT1". That makes the title read as this: Hydrogen peroxide positively regulates brassinosteroid signaling through oxidation of the transcription factor BRASSINAZOLE-RESISTANT1.

Responses to comments by reviewers

Reviewer #1 (Remarks to the Author):

This revised version of the manuscript "Hydrogen peroxide positively regulates brassinosteroid signaling through oxidation of transcription factor BRASSINAZOLE-RESISTANT1" by Bai and colleagues has been significantly edited and modified to address my previous queries and questions, as well as those of the two other reviewers.

I can only state as reviewer 1 that most of my comments were considered and addressed in a smart way, and especially the proposed additional experiments were conducted. Together with the responses to reviewers this improved the manuscript significantly. I have only a few more remarks with regard to the new data:

Response: We thank the reviewer for appreciating our effort to address all the issues previously raised. We believe that now our study is stronger and more convincing.

The procedures for the LC-MS/MS analysis and co-immunoprecipitations are not included in the M&M section.

Response: We have added procedures for the LC-MS/MS analysis and co-immunoprecipitations in Methods sections.

In Supplementary Figure 1 (b) H₂DCFDA staining for H₂O₂ in the primary root tips of Col-0 treated with BL and/or DPI. Please indicated BL+DPI picture.

Response: Thanks for pointing this, we have fixed it.

In Supplementary Figure 8 a different font is used.

Response: We have fixed it.

New references should be further considered to improve the discussion of the manuscript, especially the very related paper by Lv B, Tian H, Zhang F, Liu J, Lu S, Bai M, et al. (2018) Brassinosteroids regulate root growth by controlling reactive oxygen species homeostasis and dual effect on ethylene synthesis in Arabidopsis. PLoS Genet 14(1): e1007144.

Response: We have included this reference in our text and added the discussion as followed:

"H₂O₂ also accumulates mainly in the expanding cells of the elongation zone to promote cellular differentiation, whereas, superoxide mainly accumulates in the dividing cells of the meristem zone to maintain stem cell. BR treatment not only induces the nuclear localization of BZR1 and the accumulation of H₂O₂, but also

reduces the content of superoxide, as the BR-deficient mutant *det2-9* was recently shown to over accumulate superoxide in the meristem zone, which leads to the inhibition of root growth. Overall, the distribution pattern of H₂O₂ is similar to that of nuclear BZR1 (Supplementary Fig. 14a,b), consistent with H₂O₂ modulating BZR1 activity in root growth regulation.

This is all what I can suggest for further improvement as the authors in my opinion mostly addressed all the issues raised by the reviewers.

Response: Thanks for all excellent suggestions that will clearly improve our manuscript.

Reviewer #2 (Remarks to the Author):

1. The authors have well addressed most of my questions raised in the last review except Q4 (major), which was not sufficiently addressed. The question concerns (in transcriptome analysis) why genes involved in ROS production, photosynthesis and chloroplast were more enriched in H₂O₂-dependent BZR1-repressed genes. The authors responded that: “H₂O₂-dependent BZR1-repression genes are enriched in photosynthesis and ROS production. Such BZR1 repression of ROS-related genes may contribute to feedback regulation and redox homeostasis.” However, it is not clear from this sentence what feedback regulation will the BZR1-repressed ROS genes contribute to. ROS production? If yes, can the authors give a bit more elaboration on how BZR1 repression of photosynthesis and chloroplast related genes will contribute to this feedback regulation?

Response: Thanks for the suggestion. We have added the discussion as followed:

“H₂O₂-dependent BZR1-repression genes are enriched in photosynthesis and ROS production. BZR1 repressed the expression of several members of thioredoxins, glutaredoxins and peroxidase family genes, including *TRXf1*, *TRXh5*, *TRXm1*, *TRXm2*, *TRXm4*, *TRXy2*, *TRXz*, *GRX2*, and *PRXQ* via a H₂O₂ dependent manner (Supplementary Data 1). Such BZR1 repression of ROS-related genes may contribute to feedback regulation and redox homeostasis.”

2. Related to my minor questions #2, there are still many places in the Ms the *bzr1-1D* and *bes1-D* are not italic.

Response: We have changed them.

3. The title: A "the" should be added before "transcription factor BRASSINAZOLE-RESISTANT1". That makes the title read as this: Hydrogen peroxide positively regulates brassinosteroid signaling through oxidation of the transcription factor BRASSINAZOLE-RESISTANT1.

Response: We have changed the title to "Hydrogen peroxide positively regulates brassinosteroid signaling through oxidation of the BRASSINAZOLE-RESISTANT1 transcription factor".